**METHOD**

# AGAMEMNON: an Accurate metaGenomics And MEtatranscriptoMics quaNtificatiON analysis suite

Giorgos Skoufos[1,2,3*†], Fatemeh Almodaresi[4†], Mohsen Zakeri[4], Joseph N. Paulson[5], Rob Patro[4], Artemis G. Hatzigeorgiou[1,2,3*†] and Ioannis S. Vlachos[6,7,8*†]

* Correspondence: gskoufos@uth.gr; arhatzig@uth.gr; ivlachos@bidmc.harvard.edu
†Giorgos Skoufos, Fatemeh Almodaresi, Artemis G. Hatzigeorgiou and Ioannis S. Vlachos contributed equally to this work.
[1]Department of Electrical & Computer Engineering, University of Thessaly, 38221 Volos, Greece
[6]Cancer Research Institute | HMS Initiative for RNA Medicine | Department of Pathology, Beth Israel Deaconess Medical Center, Harvard Medical School, Boston, MA 02115, USA
Full list of author information is available at the end of the article

## Abstract

We introduce AGAMEMNON (https://github.com/ivlachos/agamemnon) for the acquisition of microbial abundances from shotgun metagenomics and metatranscriptomic samples, single-microbe sequencing experiments, or sequenced host samples. AGAMEMNON delivers accurate abundances at genus, species, and strain resolution. It incorporates a time and space-efficient indexing scheme for fast pattern matching, enabling indexing and analysis of vast datasets with widely available computational resources. Host-specific modules provide exceptional accuracy for microbial abundance quantification from tissue RNA/DNA sequencing, enabling the expansion of experiments lacking metagenomic/metatranscriptomic analyses. AGAMEMNON provides an R-Shiny application, permitting performance of investigations and visualizations from a graphics interface.

**Keywords:** Computational metagenomics, Microbiome, Quantification of microbial abundances, Identification of contaminants, Time- and space-efficient indexing/alignment

## Background

The study of metagenomics has offered us novel views of the world around us and within, while shotgun sequencing has revolutionized the field offering higher resolution and throughput [1]. Microbiome sequencing has shed light to the complex host-microbiome relationships in humans and other organisms, enabling detailed or even population-scale studies. The Human Microbiome Project (HMP) [2] and other similar studies revealed the importance of the microbiome and its implications in pathological conditions, including gastrointestinal tract inflammatory diseases, neoplastic conditions, metabolic disorders, neurodegenerative diseases, and adverse outcomes in pregnancy [3–5]. A significant source of advancements in the fields of host-microbiome interactions and microbiome manipulation for therapeutic purposes has come through the development of germ-free (GF) mice [6–9]. Utilizing GF animals, researchers can effectively transfer specific bacterial species in the gastrointestinal tract deeming these animals defined in terms of their microbiome. The field has provided hope for novel

diagnostics or even therapeutic interventions through microbiome manipulation and/or transplantation [10–12]. Currently, numerous research projects worldwide are exploring the possibilities of microbiome studies in health, disease, nutrition, reproduction, and even forensics [13, 14].

The increased resolution of shotgun DNA (metagenomics) or RNA (metatranscriptomics) samples comes along with numerous technical obstacles, mostly derived from the vast size of the generated files, reference indexes, and search space during alignment and taxonomic characterization of each sequenced read. State-of-the-art implementations have offered significant breakthroughs in most aspects of the shotgun metagenomics / metatranscriptomics pipeline, including alignment, clustering, and differential abundance analyses [15, 16]. Kraken [17], an algorithm for taxonomic label assignment, achieved a significant speed improvement by utilizing exact alignment of $k$-mers to the Lowest Common Ancestor (LCA) containing that $k$-mer. MetaPhlAn 3 [18] is a tool for profiling of microbiome communities and uses a database of unique, clade-specific gene markers. It assigns fragments by mapping them against the gene markers database using Bowtie 2 [19]. Kaiju [20] translates DNA sequencing reads into amino acid sequences and searches for maximum exact matches in a database of proteins from microbial genomes. Finally, Schaeffer et al. [21] demonstrated the ability to achieve fast and accurate read assignment transferring technology from RNA-Seq to metagenomics by applying the concept of pseudoalignment implemented in Kallisto [22], and the subsequent use of an expectation maximization algorithm for abundance estimation. Recent studies [23, 24] have shown that viral and microbial sequences can be present in host tissue and bulk cell RNA or DNA sequencing libraries. These could be the result of sample contamination, microbial infection, or local microbiota present within a sample. Leveraging processed samples could prove invaluable since they could be used for quality assessment or contaminant detection but also to quantify tissue infiltration by microbial species, viral load, or even to detect associations between host-pathogen interactions. These studies have mainly employed pipelines similar to standard RNA-Seq analyses and mostly focused on the viral content of these samples. Focusing on viral content was performed due to the lower complexity of this subproblem compared to the assignment and quantification of microbial species. Notably, a recent reanalysis of whole genome/transcriptome datasets originating from The Cancer Genome Atlas (TCGA) [25] using Kraken [17, 26] revealed microbial content in both tissue and blood samples and across different cancer types, highlighting the importance of the human microbiome for oncology-related studies using tissue-specific host samples. GATK PathSeq [27] is a recently published relevant method that aims to identify/quantify microbial sequences in DNA/RNA eukaryotic host samples and is available through the Genome Analysis Toolkit (GATK) [28]. Accurate, rapid, and memory-efficient microbial abundance quantification in host samples is still actively pursued, when the reference index comprises hundreds or thousands of microbial genomes.

We designed and implemented the Accurate metaGenomics And MEtatranscriptoMics quaNtificatiON analysis suite, AGAMEMNON [29, 30] (Figs. 1 and 2), to address still open challenges in the field but also to provide an A-to-Z approach that can support both novices and specialists. AGAMEMNON caters every step of the process, from quantification to differential abundance analysis and exploratory

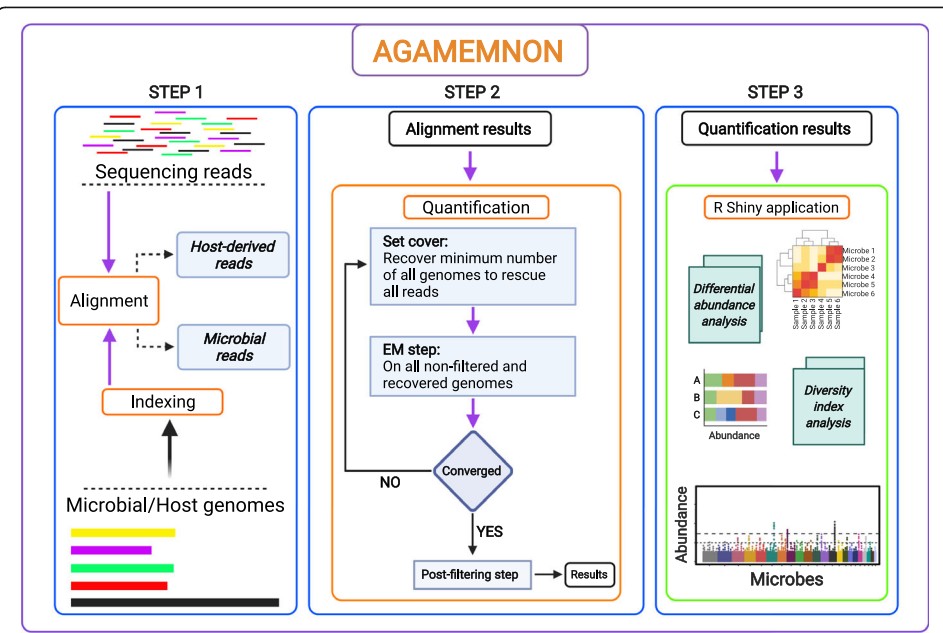

**Fig. 1** Schematic representation of AGAMEMNON. Dataset input is in raw FASTQ format. Paired-end (PE) or Single-end (SE) libraries are supported. For single-cell libraries, AGAMEMNON has helper scripts to enable per-cell analyses. In case of host tissue samples or contaminant quantification activities, the reads are first aligned against the host genome and the contaminant reference index using HISAT2. The host alignment file is saved for downstream applications and the resulting unmapped reads are forwarded to the main metagenomics/metatranscriptomics pipeline. Selective alignment is performed on the microbial reads against the reference index, while microbial abundances are subsequently quantified. A raw quantification table is produced as well as a taxonomic rank table. The results of the analysis can be used as input to AGAMEMNON's R-Shiny application, which enables diverse analyses and investigations from a graphic user interface, including visualizations, dimensionality reduction, differential abundance, and diversity index analyses

data visualization. It employs a series of advancements that enable very frugal memory requirements compared to other alignment-based approaches, enabling the use of even larger microbial indices for agnostic, hypothesis-free investigations. The pipeline for quantification follows multiple well-known existing metagenomic profiling tools [31–34], in that it divides the task of profiling into two independent steps of read alignment and subsequent abundance estimation. AGAMEMNON utilizes the Pufferfish [35] data structure for space and time-efficient representation and indexing of a collection of microbial genomes, coupled with the concept of selective alignment which allows for fast alignment of sequencing reads against a collection of genomes (Methods, "Indexing of microbial genomes"), and then feeds a novel quantification algorithm for metagenomic samples, in order to quantify the abundance of the microbial genomes. The main approach in the abundance estimation step is based on the expectation maximization (EM) algorithm and targets maximizing the likelihood of the observed reads by gradually altering the abundance value associated to different taxa. However, the EM approach is modified and adapted based on specific properties of metagenomic data; mainly, (1) high similarity among the strain sequences, (2) taxonomic tree and strain relationships through the tree hierarchy, and (3) high number of unknown species. This modified EM has been implemented as a separate module called Cedar which given the alignments from Puffaligner [36] step and reports the reference quantification at

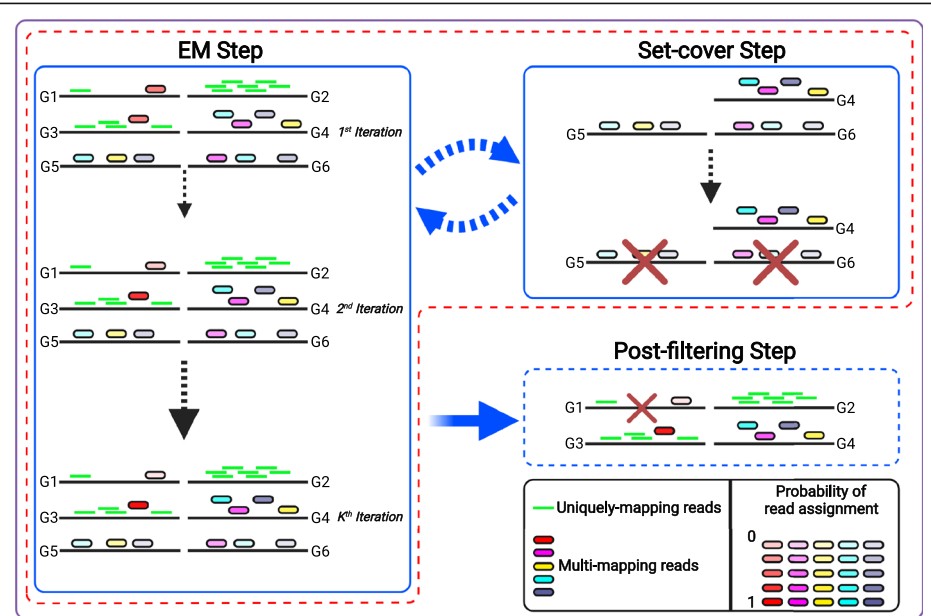

**Fig. 2** Schematic representation of AGAMEMNON's quantification engine. Each black line indicates a microbial genome. In this example, most reads are unambiguously aligned to a single genome (shown as short green lines), while 6 reads map to multiple genomes (rounded red, turquoise, purple, orange, gray, and yellow boxes). Each EM step consists of $K$ iterations (default $k = 10$). In the first EM step and first iteration, multi-mapping reads are equally partially assigned to all the genomes that they align against. For example, the turquoise read that maps to three genomes, G2, G3 and G4, is assigned a base coverage/ probability of 0.33 in each (shown by the same opacity of color in EM Step, first iteration). During EM, read assignments are resolved through iterations of reassigning the reads based on the abundance of the genomes/strains observed in the previous iteration. In each iteration, the quantification of each strain, as estimated based on the current read assignment, is used as the prior for multi-mapping read assignment in the subsequent iteration. Following each EM step (i.e., $K$ iterations), the set-cover step is also adopted, in order to resolve special multi-mapping cases that are unsolvable by the EM, called "multi-mapping islands." These are groups of highly similar strains with low abundance for which all reads are multi-mapped making it infeasible for EM to prioritize one strain over another, leading to reporting the whole group of strains with small abundances, while only few of them exist in the sample of interest, introducing false positives. The EM step - set-cover step is a looping process until set-cover is unable to remove any further genomes in which case, EM process iterates until termination. In the last step of the procedure, all the genomes with abundance values lower than a predefined cutoff are removed. In the figure's example, the process starts with six genomes (G1–G6). Throughout the iterations of the first EM step, the read probabilities change but all six genomes remain in the quantification process. When the first EM step is over, the model continues with the first set-cover step. In the set-cover step, only the genomes in which all reads are multi-mapped will be taken into consideration (i.e., G4, G5, G6). Through the set-cover process, we will keep only genome G4 and remove genomes G5 and G6 aiming for minimum number of strains that explain all multi-mapping reads. In the second EM step (not shown in the figure), only genomes G1–G4 will participate in the process. Subsequently, in this particular example, the set-cover step will never be called again because there are no multi-mapping islands left in the reference. Thus, the EM process will iterate until termination. Finally, after the whole EM process is done, the heuristic removal step will further remove the genomes whose abundance is equal to or less than 2 reads, and thus, in this example, genome G1 will also be removed before reporting the final quantification results

the requested level of the taxonomy tree. Additional modules enable concurrent deconvolution and quantification of host and microbial RNAs from the same samples or microbial abundance from host DNA samples, contaminant detection, differential abundance analysis between samples, and visual investigations using AGAMEMNON's R-Shiny [37] application. Additionally, AGAMEMNON supports single-cell techniques (host or microbial such as SiC-Seq [38]) right out of the box, for all analyses modules.

## Results and discussion

### AGAMEMNON's quantification engine

AGAMEMNON uses an expectation maximization (EM) algorithm to probabilistically resolve the origin of reads (Methods, "Quantification of microbial fragments") to individual references in the second step of the pipeline called "Cedar" (Figs. 1 and 2). This step contributes to its enhanced quantification accuracy at the species and strain levels. Unlike methods such as Kraken [17] and Kraken 2 [39] that propagate reads that have multiple best assignments to a higher taxonomic rank, AGAMEMNON, via the inference performed in Cedar, makes use of other reads and their probabilistic allocations to determine the probability that the ambiguous read arises from the different references to which it aligns well. Similar to other EM algorithms, in Cedar, we iterate over these two steps of Expectation and Maximization until the convergence criteria are met. In each iteration, we calculate the read probability distribution in the Expectation step and assign the reads across strains to maximize the probability of observed reads in the Maximization step. Based on the user's request, the same procedure can be applied at different levels of the taxonomy tree.

Cedar's EM procedure is specifically modified according to fundamental properties and challenges of metagenomic quantification. For instance, in metagenomic indexes, there is often high similarity among the strain sequences belonging to the same species [34] (sometimes even across species), which increases the complexity of disentangling reads at lower levels of the taxonomy. Additionally, reads coming from unknown species or unknown strains can be falsely assigned to entries existing in the index, resulting in false positive non-zero values. As part of the Cedar pipeline, we address these challenges through iterative, mass-preserving filtering. We look for groups of references that share the same class of reads and are fully ambiguous without any preference towards a reference over the others to detangle reads in the Maximization step. We call such a group of references, a "multi-mapping island." We reduce the problem of multi-mapping islands to a "set-cover" problem and solve it by adopting an existing approximate set-cover solution. Essentially, we select minimum number of strains that explain all multi-mapping reads distributed across the strains of each multi-mapping island. The remaining strains in each island are removed prior to the next EM step, significantly improving the accuracy of the proposed quantification model. Through this approach, we tackle the problem by sparsifying the solution (i.e., the set of species that may be assigned a non-zero abundance) in a manner that still retains all mapped reads. The "set-cover" step is called after every $k$ iterations of EM until there are no multi-mapping islands left. At this point, EM continues until termination, which happens either if (a) it reaches the maximum number of iterations (default = 1000 iterations) or (b) the genomes abundance change between the two iterations is adequately small. Cedar procedure is completed by the final step of removing genomes with abundance values lower than a cutoff threshold (default = 2).

### Comparisons with existing methods

We benchmarked AGAMEMNON using simulated [40], synthetic [41], and real datasets (Methods, "Simulated, synthetic, and real data sets") against Kaiju [20], Kraken 2 [39], Bracken [34], MetaPhlAn 3 [18], and meta-Kallisto [22] (Methods, "Benchmark

details"). Apart from Bracken and Kallisto which aim to tackle the same problem with AGAMEMNON's quantification engine and directly derive abundances from read assignment, we included in the comparisons representative algorithms from additional families of implementations, including Kraken 2, Kaiju, and MetaPhlAn. Kraken 2 (as in the case of Kraken) performs taxonomic assignment and not quantification per se, but was selected as one of the most cited metagenomic analysis methods and Kraken 2 demonstrated a series of advancements over Kraken and other methods [39]. Furthermore, Bracken uses Kraken's or Kraken's 2 results and performs a quantification step to estimate microbial abundances at the species, genus, or higher levels. Finally, Kaiju and MetaPhlAn 3 follow different approaches: DNA-to-Protein based taxonomic assignment and microbial profiling using clade-specific gene markers, respectively.

To assess the accuracy of the methods, we utilized the Mean Squared Log Error (MSLE) and the total number of reported false positive (FP) taxa in different read thresholds (Methods, "Accuracy metrics"). Shortly, MSLE is defined by the following formula:

$$L(y, \hat{y}) = \frac{1}{N} \sum_{i=0}^{N} (\log(yi + 1) - \log(\hat{y}i + 1))2$$

where $y$ and $\hat{y}$ are numeric vectors comprising the ground truth and estimated read counts respectively. $N$ is the total number of reported microbes by each method. It is practically the squared total difference between actual versus estimated log counts normalized by the total number of reported taxa. The Illumina 400 dataset [40] that was used in our benchmarks incorporates 400 different microbial genomes. A less complex version of the dataset (Illumina 100) has been commonly used as a test set in the field [21, 34, 42]. Furthermore, the synthetic dataset [41] used is a product of a real shotgun metagenomics sequencing experiment of a predefined mock microbial community. The mock community comprises 12 bacterial strains spreading over 2 phyla. Subsequently, we used seven additional samples from three studies [43–45] for which the actual bacterial abundances are known and were measured independently prior to sequencing. The choice of a reference compendium for index creation is an important aspect, since it can affect the complexity of the task at hand. To simulate real-world scenarios but also to enable us to assess the effects of reference choice on algorithm outcomes, we incorporated four different microbial references to our benchmarks. The first reference (REF-1) comprises all complete and latest bacterial and archaeal genomes from NCBI RefSeq database ($n = 1840$). In this reference, ~ 36% of the genomes present in the Illumina 400 dataset are missing and that allows us to mimic the common scenario of unknown microbial sequences in metagenomics samples. Such cases can lead to false positives, by assigning reads of unindexed microbes to the closest match in the index. The second reference (REF-2) is used for the analyses of the synthetic dataset, and it is a superset of REF-1, augmented to comprise all microbial genomes of the relevant dataset ($n = 1852$ genomes). In this reference, the ratio of present organisms vs all indexed is even smaller (12 present vs 1852 indexed). Furthermore, the third reference (REF-3) comprises 44,694 sequences (> 8500 genomes). Importantly, we removed 63% of all genomes (i.e., 252 entire genomes) from REF-3 that are part of the Illumina 400 dataset. After the aforementioned removal, reference 3 contains only 148 out of the 400

genomes that are part of the Illumina 400 dataset. Strain level results for that particular scenario are calculated by taking into account (a) how accurate the quantification of the abundances is for the 148 strains that are both part of the reference and present in the dataset and (b) how many reads are mis-classified into different strains for the 252 strains that are missing from the reference but are part of the dataset. Moreover, in the relevant test for the synthetic community, all (100%) of the strains and species present in the synthetic dataset are not included in reference 3. In that scenario, where all the strains present in the dataset are missing from the reference, we believe it is still informative to calculate metrics of accuracy. In that case, the number of the false positive strains represents the number of falsely reported taxa (in terms of presence/absence) and MSLE shows the total error in terms of mis-assigned counts (i.e., the degree of overestimated abundances for each of the falsely reported taxa). For example, a method that does not assign reads of missing strains to those present in the index will outperform a method assigning falsely the majority of those reads (while keeping all other assignments equal). Finally, the fourth reference (REF-4) is a subset of REF-3 and comprises 38,691 bacterial sequences. These tests were implemented to mimic analyses where numerous strains present in the assessed dataset are not part of the indexed annotation, which is a very common scenario especially in environmental samples. Detailed information about all three references can be found in Methods, "Benchmark details" and Additional file 4: Supplementary Table S4 - "Benchmark references".

Even though AGAMEMNON is a complete analysis suite and not just a quantification or alignment method, its engine shows robust top-of-the-line performance, across all test sets. Specifically, AGAMEMNON exhibited top performance accuracy in most of the tests. (Figs. 3, 4, and 5 and Additional file 1: Figs S1 – S7 & Figs S10, S11).

In Fig. 3, we present the results using the Illumina 400 dataset and REF-1. As shown, AGAMEMNON displayed better performance in terms of mean squared log error (MSLE) in both genus and species resolution (panels A, B) while at the strain level, AGAMEMNON and Kallisto had the lowest MSLE (panel C). MetaPhlAn 3 had the smallest number of false positives (FP) in all tested taxonomic ranks with AGAMEMNON following (panels D, E, F). Metaphlan's small number of false positives despite the low observed accuracy is expected since it uses a predefined clade-specific marker database with significantly reduced query space compared to the reference used by Kraken 2, Bracken, Kallisto, and AGAMEMNON. Kaiju was not included in the analyses presented in Fig. 3, since it only supports analyses using the complete RefSeq as reference (presented in Fig. 4) and not custom annotations.

Next, we compared the methods using both the Illumina 400 and the synthetic datasets against REF-3 index. In terms of MSLE, AGAMEMNON performed better in all tested cases and taxonomic ranks (Fig. 4, left panel). MetaPhlAn 3 had the smallest number of false positives (FP) with AGAMEMNON following (Fig. 4, right panel). Kaiju is included only in the synthetic tests, since its index (which cannot be altered) already includes the omitted species and strains. We were also not able to run metaKallisto or the REF-3, since the indexing step of REF-3 required more RAM than what was available in our largest server instance (512 GB).

To assess the concordance of the methods when analyzing shotgun metagenomics sequencing experiments, we also quantified the microbial abundances of three human stool samples originating from the Human Microbiome Project [46] using REF-3. In

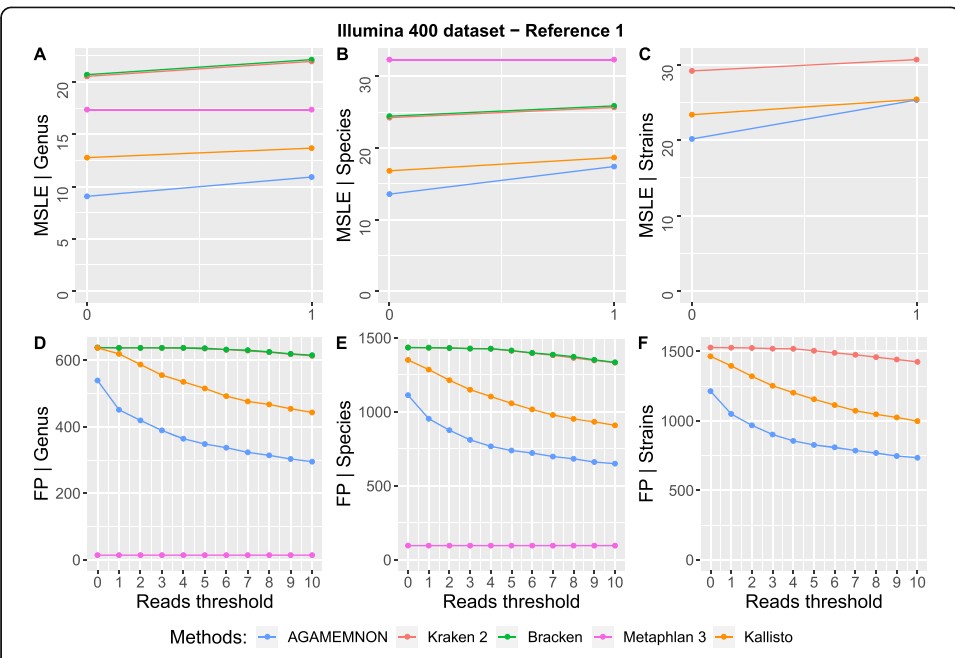

**Fig. 3 A–F** The mean squared log error (MSLE) and the number of false positive taxa (FP) between true and estimated read counts at the levels of genus, species, and strain using the Illumina 400 dataset and REF-1. We measured MSLE (a) using unfiltered results (0 x axis tick) and (b) by removing all instances where the true and estimated counts were both zero (1 x axis tick). False positive taxa were counted at all read thresholds between 0 and 10. At the read threshold of 0 reads (unfiltered results), all taxa were counted, even those with just 1 assigned read. At the read threshold of 1 read, we counted the taxa with > 1 assigned read and so on. Bracken and MetaPhlAn 3 produce results up to the species level and thus they were not included in the strain-level comparisons. Smaller MSLE and smaller numbers of false positives denote better performance

this comparison, we included Kaiju, Bracken, Kraken 2, MetaPhlAn 3, and AGAMEMNON (Fig. 5 & Additional file 1: Fig. S2). As shown in Fig. 5, all methods (excepting MetaPhlAn 3) exhibit a positive Spearman's rho > 0.5 in almost all samples and both taxonomic ranks. As expected, the highest correlation is between Kraken 2 and Bracken, since Bracken utilizes Kraken 2 output as the foundation of its abundance estimation calls. AGAMEMNON has a strong positive correlation with both Bracken and Kraken 2 at both the genus (> 0.7) and species (> 0.5) levels. These results demonstrate that most of the methods have a relative agreement in three experimentally derived human datasets. The very large number of species identified by Kaiju, Kraken 2, and Bracken (Additional file 1: Fig. 2) could be the result of an inflation due to false positives, since they are significantly larger than our current expectations for the human microbiome [46, 47]. However, since the ground truth is unknown, it is not possible to identify the most accurate approach through this evaluation alone.

In terms of execution speed and memory footprint, MetaPhlAn 3 and Kraken 2 proved to be the most efficient algorithms (Additional file 4: Supplementary Table S2 - "Execution time"). It is worth noting though that AGAMEMNON is the only method (tested in this study) that performs actual alignments against a full reference. This information (i.e., SAM files) can be stored locally and used downstream to the quantification results. The incorporation of the pufferfish data structure in the quantification engine enables AGAMEMNON to require ∼ 6.5-fold less RAM than Kallisto, a pseudoalignment-based approach. The differences are also evident during indexing, an

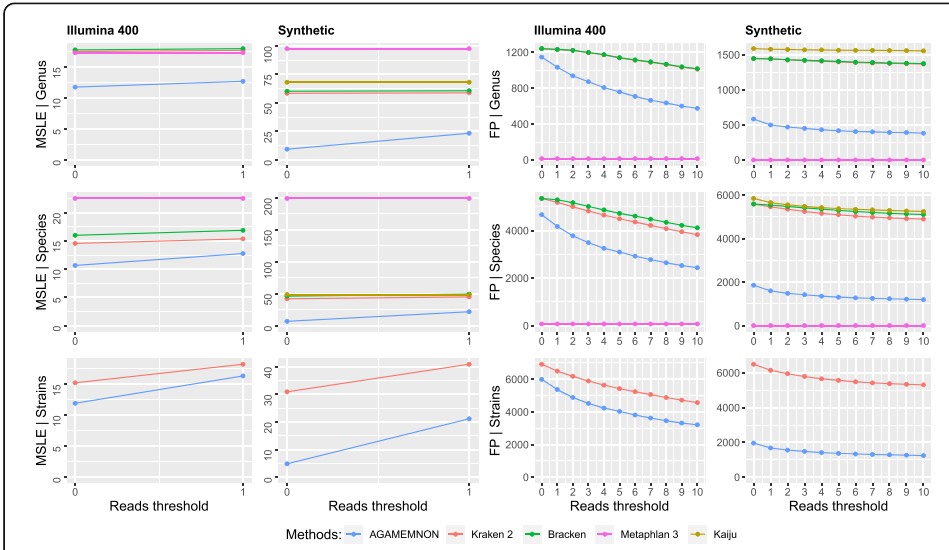

**Fig. 4** The mean squared log error (MSLE) and the number of false positive taxa (FP) between true and estimated read counts at the levels of genus, species, and strain using reference 3. We measured MSLE (a) using unfiltered results (0 *x* axis tick) and (b) by removing all instances where the true and estimated counts were both zero (1 *x* axis tick). False positive taxa were counted at all read thresholds between 0 and 10. At the read threshold of 0 reads (unfiltered results), all taxa were counted, even those with just 1 assigned read. At the read threshold of 1 read, we counted the taxa with > 1 assigned read and so on. Bracken, MetaPhlAn 3, and Kaiju produce results up to the species level and thus they were not included in the strain-level comparisons. Smaller MSLE and smaller numbers of false positives denote better performance

important bottleneck for this class of implementations, since medium to large-size microbial compendia could require more than 0.5 TB RAM for indexing, which is not always available. Specifically, for the indexing step, Schaeffer et al. [21] recommend a two-step approach; by utilizing Mash [48], the first step screens against a large collection of genomes to reduce the reference on those that have a nominal match with the sample of interest. Subsequently, Kallisto is employed for the indexing and quantification steps. We reason that sample-specific indexing can become unrealistically time- and resource-consuming but also introduce quantification biases especially in lowly abundant taxa whose genomes might be falsely ignored in some samples. We made a comparison of the execution time and memory footprint required for indexing (Additional file 4: Supplementary Table S1 - "Index benchmark") between the five methods in two different reference sizes (Additional file 4: Supplementary Table S3: "Benchmark references"): (a) REF-1 (1840 genomes) and (b) REF-3 (~ 44,000 sequences).

### Microbial quantification in host RNA/DNA-seq samples

AGAMEMNON was also designed to support the quantification of microbial fragments in host tissue/cell RNA/DNA samples. Such analyses can be useful in numerous scenarios, from contaminant identification and quantification to microbiome/viral load analysis in healthy or diseased tissues. To this end, AGAMEMNON initially separates the host sequencing reads using HISAT2 [49, 50] and subsequently employs a time and space-efficient indexing scheme to map the reads failing to align to the host genome/transcriptome against the microbial genome index. Finally, it uses its quantification engine to calculate abundances of the microbial genomes identified in the sample.

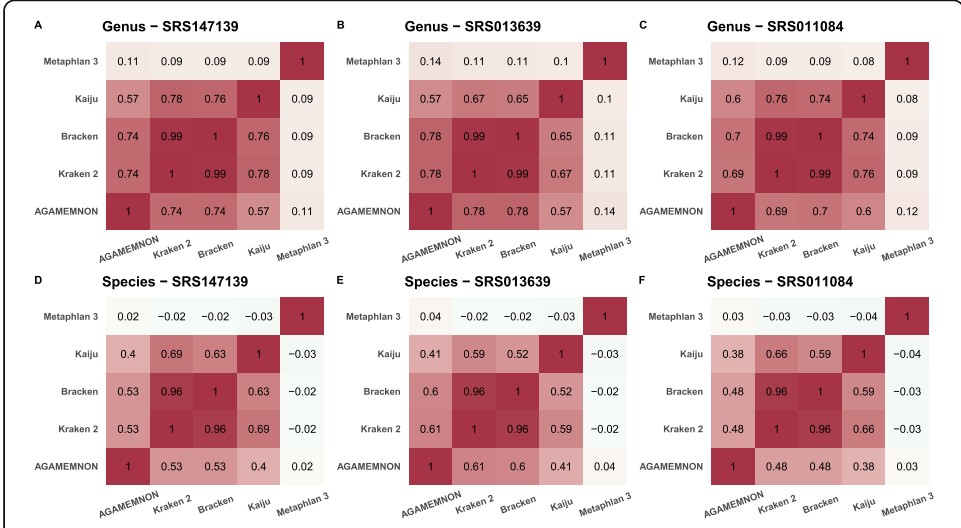

**Fig. 5 A–F** The pairwise Spearman correlation of each method in three human fecal samples at the levels of genus and species. Before calculating Spearman correlation values, we removed all instances of zero-abundant taxa from all methods

AGAMEMNON provides four ready-to-use curated microbial references (i.e., (a) Human Microbiome, (b) NCBI Complete Species, (c) Common Cell Biology Contaminants, and (d) Expanded Common Contaminants, which is an extension of (c) with viral vectors. See Methods: "Ready-to-use microbial references" for further information). Importantly, the pufferfish-based memory-efficient indexing scheme enables investigators to create their own microbial indices comprising dozens or even thousands of species without excessive RAM requirements. We evaluated AGAMEMNON's host sample analysis capabilities against GATK PathSeq [27] and the HUMAnN3 [18] pipeline (KneadData + MetaPhlAn 3) in host tissue analysis scenarios. Two different mixed simulated datasets (Methods, "Simulated, synthetic, and real data sets") were created using ART [51], which included a high (Dataset ONE, 7.53%) and a low (Dataset TWO, 3.77%) microbial read content in human. In both of the datasets and most of the taxonomic ranks, AGAMEMNON outperformed GATK PathSeq and HUMAnN3 in terms of Mean Squared Log Error (Fig. 6). HUMAnN3 had the smallest number of false positives (FP) in all tests with AGAMEMNON following. HUMAnN 3's small number of false positives despite the low observed accuracy is expected since it utilizes MetaPhlAn 3 which uses a predefined clade-specific marker database with significantly reduced query space compared to the reference used by AGAMEMNON and GATK PathSeq. Importantly, the percentage of mis-classified microbial reads to the human genome made by AGAMEMNON has no practical impact on accuracy (< 0.001%). Additionally, we calculated the number of correctly- and mis-aligned reads to the host by AGAMEMNON (HISAT2), Kneaddata (bowtie 2), and BWA which is utilized internally by GATK Pathseq (Additional file 4: Supplementary Table S14 - "Host alignments"). These results indicate that AGAMEMNON can accurately quantify microbial fragments in host-specific samples and, at the same time, separate the host sequencing reads for further analysis. This module enables users to perform such analyses in different settings, such as contaminant detection (e.g., microbes in cell cultures), local microbiota (e.g., from intestinal mucosa), or even tissue infiltrating microbes in the tumor microenvironment.

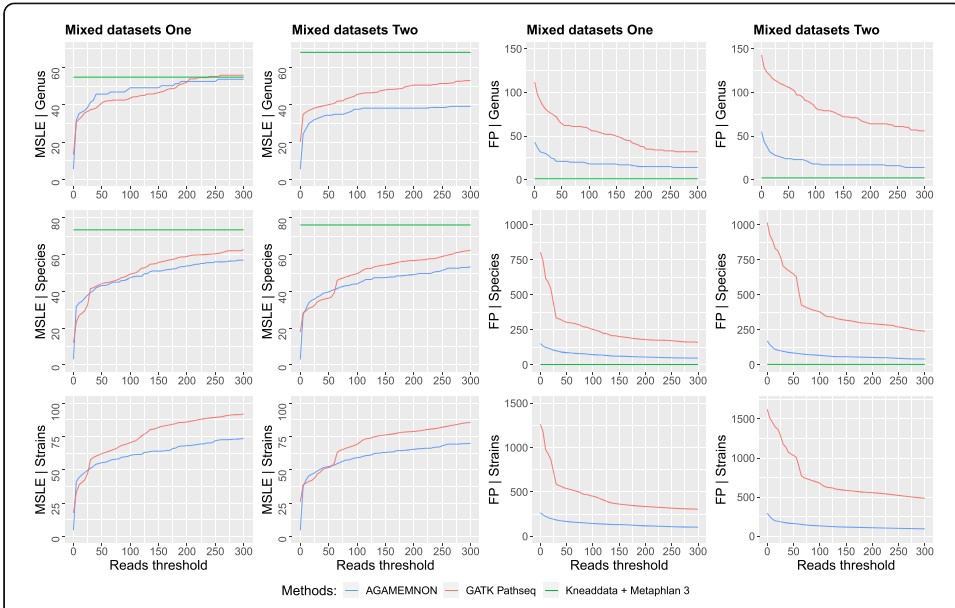

**Fig. 6** The mean squared log error (MSLE) and the number of false positive taxa (FP) between true and estimated read counts at the levels of genus, species, and strain using mixed datasets one and two and the human-subset reference. We measured MSLE and False positive taxa at read thresholds between 0 and 300 with a step of 5 reads. At the read threshold of 0 reads (unfiltered results), all taxa were counted, even those with just 1 assigned read. At the read threshold of 5 reads, we counted the taxa with > 5 assigned reads and the taxa that had < 5 reads assigned were not taken into consideration and so on. Smaller MSLE and smaller numbers of false positives denote better performance

Using 16 publicly available human datasets from the ENCODE consortium [52], we applied AGAMEMNON to identify microbial species in tissue-specific RNA-Seq samples (Methods "ENCODE data sets" and Additional file 4: Supplementary Table S6 - "ENCODE samples"). Due to its high microbial abundance and diversity [53], colon tissues are primary candidates for such analyses, since diet, infection, inflammation, and pathological conditions can alter the intestinal microbial content. Furthermore, intestinal permeability can also change, affecting the abundance and types of microbes sequenced along with the host tissue. Importantly, since microbial and host RNAs are both sequenced from the same sample, we not only get an accurate glimpse of the local microbiota but can also easily assess its effects on human tissue gene expression (and/or vice versa) or the effect of topical somatic mutations. Such localized information can be lost when using microbial sequencing data from fecal samples. The analysis of the ENCODE samples revealed a number of highly abundant bacterial species and strains, most of which are known to be abundant in the human gastrointestinal tract (Additional file 1: Figs. S8, S9).

### Downstream analyses of quantification results with R-Shiny

AGAMEMNON offers a powerful R-Shiny analysis suite (Fig. 7), where users can explore and visualize small or population-sized datasets as well as perform differential abundance or expression analyses. This module supports simple and sophisticated exploratory visualizations including heat maps, boxplots, Manhattan plots, dimensionality

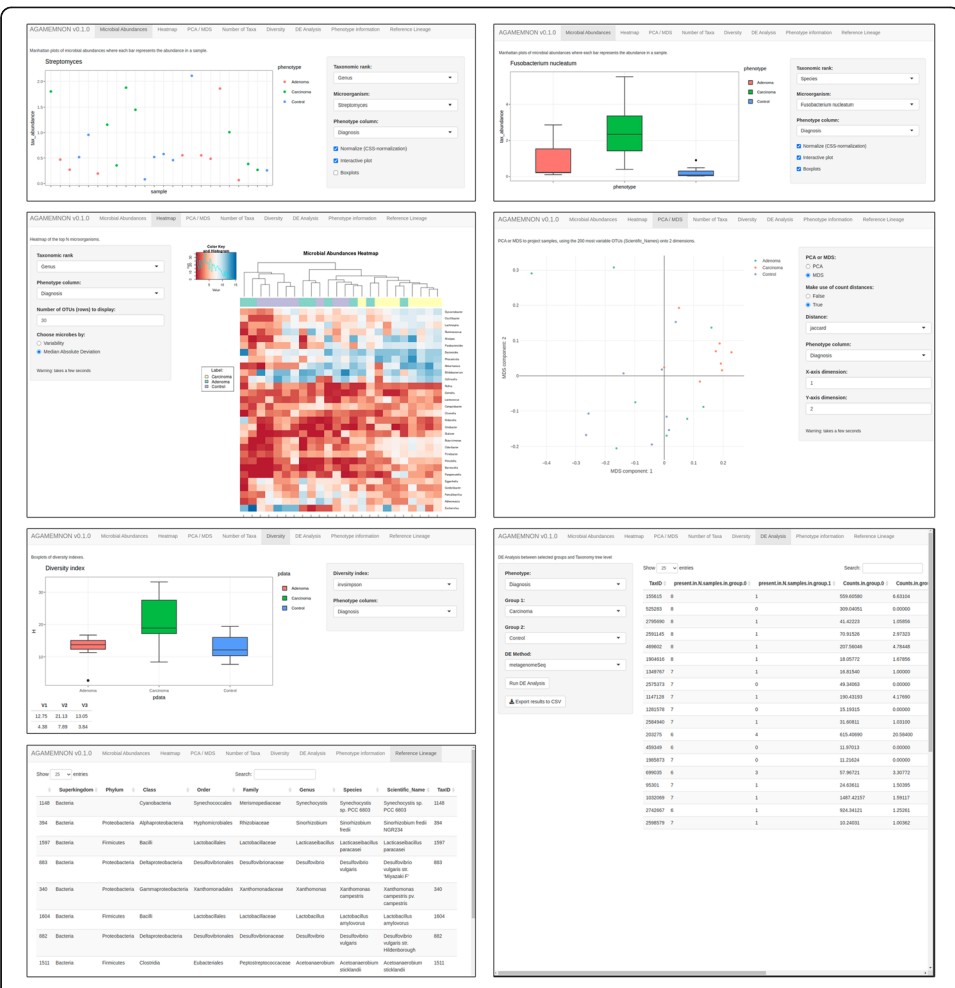

**Fig. 7** Screenshots of AGAMEMNON's Shiny application. (Top row) Visualization of microbial abundances through the use of Manhattan plots and Boxplots. (Middle row) Heatmap visualization and clustering using top N (in terms of abundance) microbes and PCA/MDS analysis. (Bottom row) Diversity index analysis and interactive tables showing the full lineage of microbes identified in the analyzed samples and differential expression analysis module and results

reduction methods (principal component analysis (PCA), multidimensional scaling (MDS)), and diversity indices (Bray-Curtis, Euclidean, Canberra). Users can interact with the application and select arbitrary phenotypic characteristics and grouping variables in order to plot and explore their effect on microbial abundances in real time. Following the grouping variable selection (e.g., healthy/patients, sex), users can also perform differential abundance (in metagenomics) / expression (in metatranscriptomics) analyses directly from the graphical user interface. Since AGAMEMNON can support a wide spectrum of research scenarios, the analysis module incorporates metagenome-specific methods, such as metagenomeSeq [15], as well as generic, and single-cell methods (i.e., limma [54], DESeq2 [55], edgeR-LRT and edgeR-QLF [56]). Finally, the application offers a series of interactive data tables presenting the phenotypic information and full lineages of the microbial genomes identified. Additional files 2 and 3 demonstrate two use-case scenarios employing visualizations and differential abundance analyses using data from the integrated Human Microbiome Project (iHMP) [57] and Feng et al. [58].

### Single-microbe sequencing analysis

We validated AGAMEMNON's single-microbe sequencing module against a single-cell artificial microbial community (Fig. 8 and Methods: "Single-cell artificial microbial community analyses") originating from a published single-cell sequencing technique, namely SiC-Seq [38]. The microbial community comprises 8 bacterial species (5 Gram positive and 3 Gram negative) and 2 yeasts. The number of sequenced cells (> 50 reads) is close to 48,000. Lan et al. [38] estimated the Read Counting values after counting cells under bright-field microscopy, and thus, we consider these relative abundances as the ground truth, as was also performed in the original manuscript. As shown in Fig. 8, AGAMEM-NON managed to accurately estimate the single-cell bacterial/yeast abundances.

### Conclusions

The decreased cost and increased availability of metagenomic and metatranscriptomic sequencing experiments have revealed the importance of the human microbiome and its role in shaping health and disease. The accurate identification and quantification of microbial abundances in such experiments are the first crucial steps in the in silico analysis of microbial communities.

To this end, we developed AGAMEMNON, a time and space-efficient in silico framework for the analysis of metagenomic/metatranscriptomic samples providing highly accurate microbial abundance estimates at genus, species, and strain resolution. Its novel indexing scheme and analysis engine enables us to go beyond taxonomic ranks with the provision of microbial abundance estimates, while bypassing the vast memory requirements of similar alignment-based quantification approaches. AGAMEMNON can index the whole human microbiome or even the complete NCBI compendium using CPU/RAM specifications available to most labs. Importantly, the employed *iterative, mass-preserving filtering* tackles effectively the very common problem of false positive

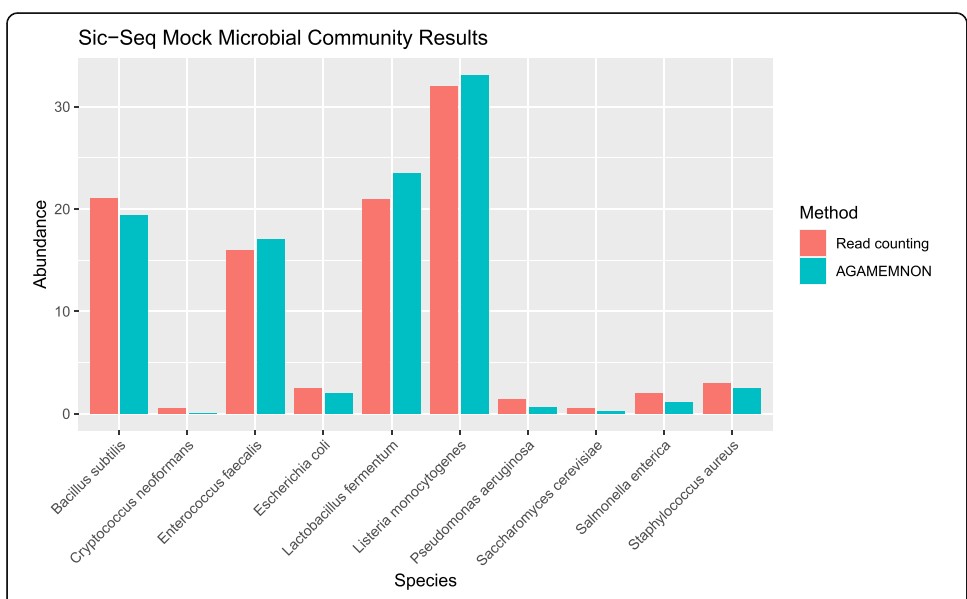

**Fig. 8** Accuracy of AGAMEMNON against a single-cell microbial community in terms of relative abundance. As stated in the Sic-Seq article, the Read Counting values emerged after counting cells under bright-field microscopy, and thus, we consider read counting as the ground truth. Microbial abundance quantification using AGAMEMNON remains highly accurate even in single-cell samples

counts in metagenomic analyses. This series of innovations enable AGAMEMNON to perform hypothesis-free quantification of diverse samples without requiring the creation of custom-tailored indexes, while exhibiting higher or equally good accuracy between all the state-of-the-art methods that were tested.

Importantly, on top of AGAMEMNON's quantification results, an R-Shiny application offers numerous downstream analyses modules that will push the envelope further, enabling users to explore and visualize microbial abundances but also conduct differential abundance and diversity index analyses through a user-friendly graphical interface.

AGAMEMNON inherently supports the analysis of single-microbe sequencing experiments, such as SiC-Seq, returning abundance estimates concordant to bright light microscopy. It also comprises additional modules, enabling the detection and quantification of contaminants as well as the streamlined extraction of microbial abundances with strain resolution directly from host tissue/body fluid RNA-Seq/DNA-Seq samples. This module provides dramatically increased accuracy compared to GATK PathSeq, enabling the acquisition of highly accurate microbial abundances from existing studies lacking a microbial arm.

In summary, AGAMEMNON improves the accuracy of microbial abundance quantification at the genus, species, and strain level, while being efficient in terms of RAM and computational time for the alignment-based class of implementations. Its best-in-class host-specific analysis capabilities and use-case versatility could enable a larger part of the community to incorporate metagenomic/metatranscriptomic investigations in their research.

## Methods

### Mapping and quantification

AGAMEMNON is implemented under a modular approach, where all modules are interconnected using the Snakemake workflow management system [59]. This approach enables users also to add functionality or substitute modules based on their needs. The mapping/quantification modules employ pufferfish to construct the microbial index using the provided reference and selective alignment to map the sequencing reads. At the quantification step, the abundance of each taxon is reported and we translate the raw results to a taxonomy-associated results file using the information provided by NCBI taxonomy database [60]. In cases where AGAMEMNON is applied to host-associated tissue/cell RNA/DNA samples, HISAT2 [50] is utilized to distinguish host/microbial reads, which are then forwarded for further analyses. After host read removal, AGAMEMNON performs a second alignment round against the bacteriophage PhiX genome which is commonly used for calibration control, as well as for color balancing and quality monitoring in Illumina sequencing. Users can extend the index of the second step, in order to add spike-ins or contaminants. After the host and sequencing control reads are removed, AGAMEMNON will continue with the identification and quantification of microbial fragments as described above using the unaligned sequencing reads.

### Indexing of microbial genomes

Pufferfish is an index based upon a compacted colored de-Bruijn graph that is able to index a large set of reference sequences efficiently [35]. Pufferfish indexes all the

subsequences of length $k$ ($k$-mers) of the reference sequences by constructing a minimum perfect hash (MPFH) [61] from the set of all $k$-mers. For each $k$-mer, Pufferfish keeps track of the $k$-mer's position in a global contig sequence, from which it is possible to recover the contig ID in which the $k$-mer appears, and its relative position on the contig. This is necessary specifically for verifying the existence of $k$-mers at the time of query, since minimal perfect hash functions do not guarantee to reject keys ($k$-mers) that were not present in the set on which they were constructed. Furthermore, for each contig, pufferfish stores the following information: the reference IDs in which the contig appears, the location of the contig in each reference, and the orientation by which the contig maps to each reference. The pufferfish index utilizes TwoPaco [62] for constructing the compacted de-Bruijn graph efficiently from the set of all reference sequences.

## Mapping of sequencing reads

The Pufferfish index enables the rapid mapping of reads from a microbial sample to a large number of genome references. First, $k$-mers from a read are queried in the index, and where they match, they are extended to maximal matches to retrieve all the locations on the compacted de-Bruijn graph where maximal matching substrings of the read exist. These matches are called Maximal Exact Matches on a unitig [63] (uni-MEMs), where a unitig is a contig subtype, defined as the sequence of non-branching paths (unipaths) in the de-Bruijn graph. Subsequently, the uni-MEMs on the compacted contigs are projected to the reference-based MEMs based on the position of contigs on the reference sequences in which they exist. On each reference, the MEMs are chained together by adopting the dynamic programming algorithm introduced in minimap2 [64] to find high scoring chains of MEMs. High scoring chains tend to include long matches between the read and the reference. Therefore, selective alignment [65] only aligns the regions between extended matches to compute the alignment scores. In this approach, a candidate search space is selected by finding the best MEM chains which are perfect matches of the read and the reference and later aligning the remaining gaps between the MEMs or at the two ends of the read to the selected reference. This hybrid approach increases the accuracy of the final alignment results while keeping the performance close to the fast and efficient exact mapping tools. The alignment of between-MEM regions is calculated by KSW2 [64, 66].

Using the alignment scores, any mapping of the reads which falls below a threshold is discarded. By default, the alignment score of a mapping should be at least 65% of the best possible alignment score for a read of the given length. This threshold allows filtering based on the quality of the match between the query and reference, so that only sufficiently high-quality matches are used for calculating the abundance of genomes. This can result in more accurate abundance estimation than methods that do not score and filter the actual alignments implied by exact matches. Furthermore, a higher alignment score indicates a higher probability of a read originating from a location, especially in the case of multi-mapping. In fact, in this model of quantification, the alignment scores are used for computing the conditional probability of a read being sequenced from a specific reference strain. Improving the abundance estimations of microbial samples by using the alignment information is also explored in other

quantification methods such as Karp [67] which also considers the base-quality information to calculate the conditional probabilities, though we did not find that information to considerably improve abundance estimation in our case (data not shown).

### Quantification of microbial fragments

We model the problem of estimating the expression of microbial strains by adopting the generative model introduced in RSEM [68] for quantification of RNA-seq reads. In that model, each fragment is generated by first selecting a reference sequence and then a position on the reference. Therefore, if we assume the same model for generating microbial reads, we can write the likelihood of observing the set of fragments given a distribution for the strain expressions as follows:

$$\mathcal{L}(\theta : \mathcal{F}) = \prod_{f_j \in \mathcal{F}} \sum_{i=1}^{M} \Pr(r_i|\theta) \; \Pr\left(f_j|r_i\right)$$

where $M$ is number of the strains, $\mathcal{F}$ is set of all fragments in the sample, and $\theta$ is the parameter showing the strains expression estimation, $\Pr(r_i|\theta)$ is the prior probability of selecting strain $r_i$ and $\Pr(f_j|r_i)$ is the conditional probability of generating fragment $f_j$ from reference $r_i$.

To compute $\Pr(f_j|r_i)$, we evaluate the compatibility of fragment $f_j$ with the reference $r_i$ by considering the alignment score computed by PuffAligner [69]. Furthermore, we also consider the effective length and coverage ratio of the reference for computing the prior probability of selecting a strain to be sequenced.

In this step of the pipeline, we use the expectation maximization (EM) algorithm for estimating $\theta$, abundances at the level of strains. We apply this iterative algorithm over a reduced representation of the data provided by rich equivalence classes [70] until the parameter estimates converge or we reach the maximum allowed number of iterations. This reduction makes the optimization of the objective function practically tractable by decreasing the amount of computation for the large space of fragment associated variables. As per definition, two fragments $f_i$ and $f_j$ are equivalent if they align to the exact same set of references, in which case, they belong to the equivalence class that is labeled by those references. In the new optimization process, the likelihood objective function that is optimized is the probability of observing the set of equivalence classes rather than read fragments with the frequency of each equivalence class defined by total number of reads belonging to that class. Subsequently, we adjust the objective function for different biases in the data such as variation of coverage for references belonging to the same equivalence class, an identical process to the one followed in Salmon [70], as well as by applying the range factorization improvement to break the equivalence classes into more fine-tuned and accurate approximations of the fragment distributions without the loss of tractability and performance, as explained in Zakeri et al. [71]. The likelihood function after representing all the fragments as range factorized equivalent classes is:

$$\mathcal{L}(\theta : \mathcal{F}) \sim \prod_{\mathcal{F}^q \in C} \left( \sum_{r_i \in \Omega(\mathcal{F}^q)} \Pr(r_i|\theta) \; \Pr(f|\mathcal{F}^q, r_i) \right)^{N^q}$$

where $C$ is the set of all equivalent classes, $\Omega(\mathcal{F}^q)$ is the set of all strains in equivalent class $\mathcal{F}^q$, $N^q$ is the number of fragments in the equivalent class $\mathcal{F}^q$, and $(f|\mathcal{F}^q, r_i)$ is

the conditional probability of observing a fragment from strain $r_i$ in equivalent class $\mathcal{F}^q$. As shown by the equation, we treat all the $N^q$ fragments in the equivalent class $\mathcal{F}^q$ equally and assign the same conditional probabilities to them given each strain. Summarizing the fragments into equivalent classes reduces the number of updates required in each EM iteration and therefore improves the efficiency of the EM algorithm.

In many metagenomic quantification tasks, considering the depth of the high-throughput sequencing techniques and the large amount of sequence similarity across genomes, low abundance sequences or taxa are usually discarded from the quantification results. One solution proposed to avoid reporting a large number of false positive genomes is having a post-filtering step to discard references with abundances smaller than a predefined cutoff value or discard all the multi-mapped reads [34, 67]. This post-filtering results in losing the reads that were mapped to those discarded references. However, in our pipeline, rather than discarding these reads, we keep and distribute them among the remaining references. This can help better estimate the counts of the remaining references based on the following principles:

1. Each read is supposed to be the output of a sequencing process and contains information, so discarding the read implies losing information.
2. Each read can be multi-mapped to the true origin and some other references in case of no errors in the sequencing.
3. The sequencing error rate is low and if the mapping tool allows sub-optimal mappings, we can rely on the fact that we almost always have the true genome of origin among the set of references for a multi-mapped read.

Considering these observations, we augment the EM algorithm with an iterative, mass-preserving thresholding step.

Iterative thresholding is the solution we propose to tackle the problem of sparsifying the abundance report at which we arrive without the loss of reads. Throughout the EM process, at the beginning of every $k$ iterations (default $k = 10$), we go over the following four steps of thresholding:

1. Mark references as potentially removable (PR): We examine all references and mark a reference as potentially removable if it has a count smaller than or equal to the given cutoff. Our main goal is to discard as many references as possible from the list of potentially removable references without losing any reads.
2. Remove safe PR references immediately: A read will be lost if all the references that are equivalent over this read are discarded as the equivalence class that the read belongs to will contain no remaining references. Therefore, out of the list of the PR references, those for which there is a non-PR reference belonging to each equivalence class in which this reference appears in the label can be safely removed.
3. Solve the set-cover problem for unsafe PR references: For any reference $g_i$ in the remaining set of PR references, there exists at least one equivalence class such that all of its references are PR, including $g_i$. We call such equivalence classes critical equivalence classes, and according to our goal, we want to determine a minimum

number of PR references that can cover all critical equivalence classes, while still assuring that each aligned read can still be allocated to a retained reference. The problem can be easily reduced to set-cover. Each reference represents a set of critical equivalence classes (those that it is part of), and we want to select the minimum number of sets (i.e., references) that can explain all the set members (i.e., critical equivalence classes). Set-cover is NP-Hard [72]; therefore, we employ the greedy approximation algorithm to obtain a set of retained references. The remaining elements (i.e., genomes) at the end of the set-cover process are those that cannot be removed. For the rest of the list, we can safely remove them without losing any reads.

4. Update equivalence classes: In the last step, we need to update all equivalence classes that have lost any reference and the weights of the references for the next iterations of EM.

We would like to mention that even tough modifying the list of genomes through iterative thresholding introduces a new likelihood function at each EM step, the EM algorithm's termination is still guaranteed. When we perform a set-cover step, either it is idempotent, and so the set of references remains the same, and the termination of the procedure follows from the termination of the EM, or we remove at least one reference. But we can remove a reference at most $N$ times ($N$ = number of references). So, this procedure must always terminate.

### Simulated, synthetic, and real data sets

To evaluate the accuracy of AGAMEMNON in a setting where the true microbial abundance levels are known, we used both a simulated [40] and a synthetic dataset [41] of a predefined mock microbial community (mock), but also constructed two mixed host-microbiome simulated data sets using ART [51]. Moreover, we used three shotgun metagenomics sequencing experiments from human stool samples. Finally, we used seven real samples from three [43–45] additional studies for which the actual bacterial abundances were measured independently prior to sequencing. The simulated dataset Illumina 400 is a publicly available complex simulated microbial community comprising more than 240 microbial genera belonging to more than 350 different species scattered in 400 sub-species/strains. It consists of ∼ 20 million paired-end reads with an average read length of 75 bp. The synthetic dataset is publicly available and produced by conducting a real shotgun metagenomics sequencing experiment in a predefined mock microbial community. It comprises 12 bacterial strains belonging to 2 phyla. It consists of 215 million paired-end reads with a read length of 150 bp.

We also constructed two mixed host-microbiome simulated data sets comprising reads originating from the human genome (GRCh38 primary assembly), and 18 bacterial genomes. The two simulated data sets comprise ∼ 64 million and ∼ 67 million, paired-end reads, respectively, with a read length of 125 bp. The fragment mean size was set to 300 bp and the standard deviation of fragments was set to 50 bp. Finally, in order to simulate the sequencing error and the per base-quality scores, we used an empirical-error model of the Illumina HiSeq 2500 system, as implemented in ART [73]. The fraction of microbial reads in relation to the total number of reads for each of the

two data sets, is equal to 7.53% and 3.77%, respectively. On a real experiment, we expect sequencing reads to appear in random order, and thus, all of the simulated datasets above were randomly shuffled once prior to benchmarking. Finally, we quality-checked, pre-processed, and used three real shotgun metagenomics sequencing samples (with unknown abundances) to assess the potentiality of the methods in experimentally derived, human gut datasets and seven real shotgun metagenomics sequencing samples with known bacterial abundances. After the pre-processing step, the three samples comprise ~ 111 million, ~ 82 million, and ~ 92 million, paired-end reads, respectively, with an average read length of 101 bp. The seven (paired-end) samples comprise ~ 4.M, ~ 6 M, ~ 5.4 M, ~ 19 M, ~ 26 M, ~ 2.5 M, and ~ 5.4 M reads respectively. The range of their read length is between 40 and 150. Their accession numbers are listed in Additional file 4: Supplementary Table S9 - "BENCHMARK-Real samples."

### Benchmark details

We benchmarked AGAMEMNON against MetaPhlAn 3 [18], Kraken 2 [39], Bracken [34], Kallisto [21, 22], and Kaiju [20] using simulated, synthetic, and real datasets. For the simulated dataset [40], we mapped the sequencing reads against two references (REF-1 and REF-3, Additional file 4: Supplementary Table S3 - "Benchmark references") for Kraken 2, Bracken, and AGAMEMNON. We were not able to run meta-Kallisto in the Illumina 400/REF-3 test since the indexing step of REF-3 required more RAM than what was available in our largest server instance (512 GB), and thus, it was only included in the Illumina 400/REF-1 scenario. Also, Kaiju was not considered at the Illumina 400 dataset comparisons since its index already includes the omitted genomes. The REF-1 reference incorporates all of the representative, reference bacterial, and archaeal genomes from NCBI RefSeq. In this reference, ~ 36% (143 out of 400) of the microbial genomes present in the simulated dataset are missing. The total number of complete genomes for REF-1 is 1840. The REF-3 reference incorporates all of the complete/assembled microbial genomes from NCBI RefSeq (44,694 sequences, ~ 8600 genomes). Importantly, 63% of all genomes (i.e., 252 entire genomes) from reference 3 that are part of the Illumina 400 simulated dataset have been removed. After the aforementioned removal, reference 3 contains only 148 out of the 400 genomes that are part of the Illumina 400 dataset. Furthermore, 100% of all genomes and species present in the synthetic dataset (i.e., 12 out of 12 strains/species) are not included in the REF-3 index.

For the synthetic dataset [41], we used two references (REF-2, REF-3, Additional file 4: Supplementary Table S3 - "Benchmark references") for Kraken 2, Bracken, and AGAMEMNON. For the aforementioned reasons, meta-Kallisto was not included in the Synthetic dataset/REF-3 test and Kaiju was not considered in the Synthetic/REF-2 scenario. The REF-2 reference incorporates all of the representative, reference bacterial, and archaeal genomes from NCBI RefSeq and all microbial genomes present in the synthetic dataset.

For the seven real samples with known abundances, we used REF-3 and REF-4. Reference 3 was used against all seven samples while REF-4, which is a subset of REF-3, was used on one (SRR2726667). REF-4 was constructed by completely removing all the genomes belonging to 9 species present in sample SRR2726667. In these comparisons, we were not able to run meta-Kallisto since the indexing step for REFs 3 and 4 required

more RAM than what was available in our largest server instance (512 GB). Kaiju was not considered in the REF-4—sample SRR2726667 comparison since its index already includes the omitted species.

Prior to benchmarking, we quality-checked and pre-processed the simulated, synthetic, and real datasets using FastQC [74] and cutadapt [75]. We mainly focused on the trimming of low-quality bases (i.e., trimming bases with a Phred score < 10 and keeping only reads with minimum length > 35 bp).

For all accuracy tests mentioned above, we used the Mean Squared Log Error (MSLE) accuracy metric and the number of false positive taxa identified by each method. In order to calculate the metrics between the true and estimated counts and to produce the manuscript plots, we used R versions 3.6 and 4.0.

Finally, we benchmarked AGAMEMNON against GATK PathSeq [27] and HUMAnN3 (KneadData + MetaPhlAn 3) using two mixed host-microbiome simulated datasets. GATK PathSeq's output contains (a) read counts for all of the TaxID's (strains, sub-species) with at least 1 assigned read and (b) all other taxonomic ranks (species, genus, family, etc.) up to the root node of the lineage by summing up read counts of the lower taxa. In order to calculate GATK PathSeq's results at the TaxID level (strains, sub-species), we selected all common TaxID's between the reference, GATK PathSeq's output, and AGAMEMNON's output and for all of the higher taxonomic ranks, we grouped them appropriately and summed them up until we reach the rank of interest. In addition, for every taxon, GATK PathSeq contains two read count fields, the first refers to unambiguous reads and the second to all reads (ambiguous, unambiguous). In the results, the "reads" field is reported, since it refers to all reads, comprising uniquely and multi-mapped reads. For all three methods, in all scenarios, we used the GRCh38 primary assembly genome and a subset of the human-specific reference (human-subset, Additional file 4: Supplementary Table S3 - "Benchmark references") to map the host's and microbiome reads respectively. We measured the execution time and peak RAM memory using Linux bin/time. All tests were performed on a CentOS Linux release 7.5.1804 server with two Intel Xeon processors (12 cores each, 48 threads total) and 512 GB of RAM.

**Accuracy metrics**

The mean squared log error (MSLE) was computed using the following formula:

$$L(y, \hat{y}) = \frac{1}{N} \sum_{i=0}^{N} (\log(yi + 1) - \log(\hat{y}i + 1))2$$

where $y$ and $\hat{y}$ are numeric vectors comprising the ground truth and estimated read counts respectively. $N$ is the total number of reported microbes by each method. In Figs. 3 and 4 and Additional file 1: Fig. S1 and Figs. S3-S7, we first calculated MSLE (a) using results without any filtering (0 $x$ axis tick) and (b) by removing all microbial instances where the true and estimated read counts were both zero (1 $x$ axis tick). The filtering step conducted in (b) prior to MSLE calculation, affects the $y$ and $\hat{y}$ vectors (i.e., it removes the values that are zero-abundant in both the ground truth and the estimated counts from the vectors) but also changes $N$ (i.e., it reduces it to the new total number of observations after removal of the zero-abundant taxa). In Fig. 6, the same

procedure was followed, but this time we started by using results without any filtering up to the threshold of 300 reads using 5 read step increments. For instance, when the read threshold is 25, all microbial taxa with < 25 assigned reads are removed from both $y$ and $\hat{y}$ vectors and MSLE is re-calculated with the updated vectors and the updated $N$ value.

The number of false positive taxa was counted for sequential read thresholds between 0 and 10 (1 read step increment) for Figs. 3 and 4 and Additional file 1: Fig. S1 and Figs. S3-S7 as well as between 0 and 300 (5 read step increment) for Fig. 6. At the zero read threshold (no filtering), all false positive taxa were counted, even those with only 1 read assigned to them. As the read threshold increases, only the taxa reaching and exceeding it are taken into account.

### ENCODE data sets

The analysis presented in Additional file 1: Figs S8, S9 was performed using 16 samples originating from Peyer's patch, sigmoid colon, stomach, and transverse colon tissues from publicly available ENCODE data. The accession numbers and associated information are listed in Additional file 4: Supplementary Table S6 - "ENCODE samples." All samples were analyzed using AGAMEMNON's host-associated tissue-specific mode. Since we analyzed human samples, we used AGAMEMNON's ready-to-use human microbial reference. All sequencing datasets have been produced by the same laboratory (Thomas Gingeras Lab, CSHL). As stated in the protocol section of the sequencing experiments, Ambion mix 1 spike-ins were utilized for sequencing control, and thus, the second alignment round of AGAMEMNON was used to remove these spike-ins and not the default PhiX. All utilized ENCODE data sets were quality-checked and pre-processed using FastQC [74] and cutadapt [75] once prior to the analysis.

### Single-cell artificial microbial community analyses

We used AGAMEMNON to analyze an artificial microbial community originating from a microbial single-cell sequencing technique (SiC-Seq) [38]. The community consists of 8 bacterial species and 2 yeasts. We created a reference comprising 116 complete genomes (278 sequences) including the genomes used to create the synthetic community. The rest of the genomes were microorganisms belonging to the same genera as the species used for the construction of the synthetic microbial community (Additional file 4: Supplementary Table S5 - "single-cell genomes"). We quality-checked and pre-processed the SiC-Seq dataset by grouping the reads under the barcode they belong and removed all barcodes with < 50 reads, yielding a little more than 48,000 unique barcode groups (cells). We then applied AGAMEMNON to identify and quantify the abundances of the microbial species. Finally, similarly to the analyses conducted in Lan et al. [38], we considered the most commonly mapped species (in terms of read counts) as the species from which the cell originates. We then counted the number of cells per species, divided with the total number of cells and multiplied by 100 to acquire the species' relative abundances. We utilized the bright-field microscopy results from SiC-Seq paper as the reference annotation for this dataset and to derive AGAMENON's accuracy in single-cell data.

### Ready-to-use microbial references

AGAMEMNON supports a wide variety of research settings including investigations of microbial abundance (Genus, Species, sub-Species) in host-derived samples (tissue/cell RNA/DNA, fecal). Therefore, we created 4 distinct indices to support different analysis scenarios: (A) Human microbiome: a manually curated dataset for use in human-specific investigations. This human-specific reference comprises ~ 2233 bacterial and 10 archaeal genomes, mainly retrieved from the Human Microbiome Project [2] and a manually curated corpus of independent studies which comprised samples from different human body sites (Additional file 4: Supplementary Table S4 - "Ready-to-use references"). (B) NCBI Complete Species: the second ready-to-use reference has a more generic nature, comprising all of the NCBI's complete species and reference bacterial genomes, amounting to ~ 1687 bacterial genomes, 138 reference archaeal genomes, and 17 eukaryotic genomes (Additional file 4: Supplementary Table S4 - "Ready-to-use references"). (C) Common Cell Biology Contaminants: the third reference contains ~ 120 common contaminant microbial organisms derived from curating the literature to support analyses where AGAMEMNON is used for contaminant detection (Additional file 4: Supplementary Table S4 - "Ready-to-use references"). (D) Expanded Common Contaminants: Finally, the fourth and final reference consists of the dataset C), expanded with ~ 1300 viral vector sequences (Additional file 4: Supplementary Table S4 - "Ready-to-use references").

### Visualizations and data exploration modules

Using R-Shiny, we developed a graphical user interface for the interactive visualization and exploration of the results produced by AGAMEMNON (Fig. 7). Apart from the visualizations and exploratory modes, AGAMEMNON also hosts a set of meta-analysis modules that can be executed on-the-fly. The application supports heatmaps, boxplots, and Manhattan plots of the microbial abundances and/or expression, taxa identified, and the detection rates per sample, or any other phenotypic characteristic upon selection. Users can select grouping variables and perform exploratory investigations in real time. The meta-analysis modules include diversity index analyses (Shannon, Simpson), principal component analysis (PCA), and multidimensional scaling (MDS) while supporting between different types of distances (Bray-Curtis, Euclidean, Canberra, etc.). Importantly, users can perform differential abundance/expression analyses using different models and methods directly from AGAMEMNON. Finally, the application offers a series of interactive data tables containing the phenotypic information and full lineages of the microbial genomes identified. Under the hood, our Shiny application is developed in R using a series of R libraries including Shiny, metagenomeSeq, and biomformat [76].

### Differential abundance/expression analysis statistical models

The differential abundance and expression analysis module is implemented in R and designed to support a wide spectrum of research settings and scenarios. The default method is the Zero-Inflated Log-Normal distribution mixture-model with posterior probability weighing from metagenomeSeq package (fitFeatureModel) [15]. AGAMEMNON also supports limma with mean-variance modelling at the observational level

(voom) for generalized linear model analyses, as well as limma with precision weights per sample to account for variations in precision between different observations [77]. DESeq2 [55] has been incorporated as a robust negative-binomial distribution-based method. For single cell or datasets with high numbers of dropouts or sparsity, we implemented the likelihood ratio test (LRT) or quasi-likelihood F-tests (QLF) from the edgeR package (in R) which have been shown as quite robust in such scenarios [56].

### Single-cell metagenomics analysis

The single-cell module is responsible to split reads into read groups (e.g., cells or wells depending on the library preparation) and to keep track of them across the mapping and quantification engine. Users can provide read structure and/or an index table (depending on the prep). Due to the nature of the single-cell library-prep protocols, there is a strong possibility that some cells will end up with a small number of sequencing reads. Users can select a threshold parameter for cells to be forwarded to downstream analyses and visualizations. AGAMEMNON takes into account the sparsity of single-cell approaches also in the differential abundance analyses, whereas mentioned in the relevant section above, it offers methods shown to be robust in such settings such as edgeR-LRT and edgeR-QLF [56].

### Software versions, options, data acquisition

All tests were performed with Kaiju v1.8, MetaPhlAn v3, Kallisto v0.44.0, Kraken 2 v2.0.8, Bracken v2.0 GATK PathSeq, KneadData, and AGAMEMNON-v0.1.0. All of the methods, including AGAMEMNON, were run with default parameters. Genomes were downloaded from NCBI on Nov 27 2019 (for REF-1 and REF-2) and on May 4 2021 (for REF-3, REF-4).

## Supplementary Information

---

**Additional file 1: Supplementary figures S1 – S11.** Figures demonstrating benchmark results between the methods compared throughout the study. AGAMEMNON results using 16 publicly available RNA-Seq tissue samples from the ENCODE consortium.

**Additional file 2.** Demonstration of the first use-case scenario using AGAMEMNON and employing visualizations and differential abundance analyses using data from the integrated Human Microbiome Project.

**Additional file 3.** Demonstration of the first use-case scenario using AGAMEMNON and employing visualizations and differential abundance analyses using data from the Feng et al.

**Additional file 4.** Supplementary tables.

**Additional file 5.** Review history.

---

#### Acknowledgements

The authors would like to thank the Human Microbiome Project and the ENCODE Consortium, as well as the respective laboratories for producing data used in this project. Portions of this research were conducted on the O2 High Performance Compute Cluster, supported by the Research Computing Group, at Harvard Medical School. The research was also supported computationally by the Ithaca High Performance Compute Cluster, BIDMC. The authors would also like to thank Dr. Spyros Tastsoglou for the helpful discussions about the manuscript.

#### Review history

The review history is available as Additional file 5.

#### Peer review information

**Authors' contributions**
RP, GS, and ISV designed the method. GS, FA, MZ, JNP, RP, and ISV designed and implemented the algorithms, the pipeline, and conducted analyses. RP, AGH, and ISV supervised the study. All authors contributed in drafting and revising the manuscript.

**Author's information**
Twitter handle: @ioavlachos (Ioannis S Vlachos).

**Funding**
ISV has been supported by the "George and Marie Vergottis" foundation and the Harvard Medical School Initiative for RNA Medicine Pilot Grant Program and the Harvard Medical School Initiative for RNA Medicine (HIRM) Pilot Grants Program. GS is supported by the Operational Programme "Human Resources Development, Education and Lifelong Learning" in the context of the project "Strengthening Human Resources Research Potential via Doctorate Research" [MIS-5000432], implemented by the State Scholarships Foundation (IKY), in the form of a PhD Scholarship. MZ, FA, and RP are supported by NIH R01 HG009937 and by National Science Foundation awards CCF-1750472 and CNS-1763680. AGH has been supported by "ELIXIR-GR: The Greek Research Infrastructure for Data Management and Analysis in Life Sciences" [MIS-5002780], implemented under the Action "Reinforcement of the Research and Innovation Infrastructure," funded by the Operational Programme "Competitiveness, Entrepreneurship and Innovation" [NSRF 2014–2020] and co-financed by Greece and the European Union (European Regional Development Fund).

**Availability of data and materials**
AGAMEMNON is released under the MIT License and is available at: https://github.com/ivlachos/agamemnon [79] and also at zenodo [80]. The scripts we used to calculate metrics and create the figures of the manuscript are available at: https://github.com/gskoufos/AGAMEMNON-manuscript [81].
The datasets used during this study are included in the following published articles and open access repositories: Sevim V et al. [41], Lan F et al. [38], Qiang Feng et al. [58], Aaron M. Walsh et al. [45], Wenyi Xu et al. [44], Marcus B. Jones et al. [43], https://www.encodeproject.org/ [52], and https://hmpdacc.org/ [78].

# Declarations

**Ethics approval and consent to participate**
Not applicable.

**Consent for publication**
Not applicable.

**Competing interests**
RP is a cofounder of Ocean Genomics Inc. The remaining authors declare that they have no competing interests.

**Author details**
[1]Department of Electrical & Computer Engineering, University of Thessaly, 38221 Volos, Greece. [2]Hellenic Pasteur Institute, 11521 Athens, Greece. [3]DIANA-Lab, Department of Computer Science and Biomedical Informatics, Univ. of Thessaly, 351 31 Lamia, Greece. [4]Department of Computer Science, University of Maryland, College Park, MD, USA. [5]Department of Data Sciences, Genentech Inc., South San Francisco, CA, USA. [6]Cancer Research Institute | HMS Initiative for RNA Medicine | Department of Pathology, Beth Israel Deaconess Medical Center, Harvard Medical School, Boston, MA 02115, USA. [7]Spatial Technologies Unit, Beth Israel Deaconess Medical Center, MA, Boston, USA. [8]Broad Institute of MIT and Harvard, Cambridge, MA 02142, USA.

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
