## [**Additional file 5.** Review history. · Genome Biology]

Review History

First round of review

Reviewer 1

Are you able to assess all statistics in the manuscript, including the appropriateness of statistical tests used? No.

Comments to author:

Skoufos et al. developed AGAMEMNON a method for metagenomic and metatranscriptomic quantification. Short reads are searched against a pufferfish index, which stores the reference genomes and queries them. Once seeds are found they are chained and scored by an alignment. Using these hits, a maximization expectation algorithm estimates the abundance of each species in the sample. Additionally, to the quantification method, the authors developed a visualization in R shiny.

Major:

- I tried to run the software on my MacOS notebook, however pre-compiled binaries are only provided for Linux. Please state what operating systems are supported. I could not find the required software versions of HISAT2, Pufferfish (and cedar?) to manually install so I could still try out the software. Please provide a list of versions of the software dependencies. Additionally, I would strongly recommend to package the software in bioconda to make it easy for users to install.

- What is cedar (binaries/cedar)? Is that a novel part of this manuscript? If so, where is the source code?

- The authors mention that a biased reference can cause mis-quantification. However, in the benchmark only 1,840 reference genomes were used. This set is too small considering that GenBank contains hundreds of thousands of genomes. This reference set is also too clean since many draft genomes contain contamination, redundancy and are fragmented. From the current benchmark it is not clear to me how this method would perform on data like this.

Users who work with environmental microbial data (e.g. soil) may want to use GenBank as a reference. If the software does not perform well on this type of data, please declare it.

Please add a benchmark considering this attribute.

- In figure 2 and 3 it is not intuitive that the MSLE increases with higher read threshold while false positive number decreases. How are unclassified queries labelled? It would be better to separate how many queries are classified and how accurate they are classified. Please explain in detail how the MSLE was computed.

Minor

- Please open-source the scripts to reproduce the results
- Table 1 states that the benchmark set "illumina 400" has ~20M reads. However, on page 6 line 33 it is mentioned that it consists of 53.3 million paired-end reads. Does the QC of the reads remove 23.3 million reads? If yes, why is the QC that strict?
- Kraken2 seems to be consistently 3-4 times faster than AGAMEMNON, except in the rep-ref-ncbi-mock benchmark. Why is AGAMEMNON so much faster here? The size of the database seems to be about equal.
- Adding a link to the software in the abstract would make it easier to find the software
- line 52 p 12. "Encode samples" should be "ENCODE samples"
- The name of x-axis should consistent between figure 2 and figure 3
- In figure 4, "The six most abundant species" is written in the legend. However, there are more than six species in the graph
- Please consider uploading all code in the git repository and not only providing it as part of the GitHub Released versions.

Reviewer 2

Are you able to assess all statistics in the manuscript, including the appropriateness of statistical tests used? No.

Comments to author:

This paper describes AGAMEMNON, a software pipeline that predicts the relative abundance of taxa in metagenomic or metatranscriptomic samples. In microbiome research we are seeing more of a shift away from correlative studies to more causative studies, driven by the application of metagenomics and/or metatranscriptomics. Currently there is a need for pipelines that are able to process these datasets. While most interest focuses on functions captured by these datasets, there is still interest in assigning taxonomic labels to the sequences. At present there are a number of tools that seek to assign taxonomic labels to these datasets, including some that the authors have benchmarked their tool against here. There are also pipelines that take the user through the various sequence processing steps - notably HUMAnN2 (now HUMAnN3) which appears to do a similar analysis as AGAMEMNON (albeit it also predicts microbiome function). The main advance as far as this reviewer understands, is the application of an expectation maximization algorithm to predict the taxa present in a sample. While algorithms such as KRAKEN may assign many thousands of taxonomic labels to a dataset, AGAMEMNON takes the approach that many of those labels may be false positives and instead attempts to assign sequences to a more limited set of 'likely' taxa, which seems a similar idea to MetaPhlan, although through a different

method. That said, the paper is presented in a way that makes it difficult for the reader to understand what the authors are attempting to present. On the one hand there is a new algorithm for assigning taxonomic labels, on the other there is a sequence processing pipeline. Both aspects seem to have got obscured in the writing, such that it is not clear what the authors are presenting.

I would recommend that the authors consider rewriting the manuscript, that more clearly describes AGAMEMNON as a software pipeline for processing metagenomic/metatranscriptomic datasets. The EM algorithm is also buried in the methods and not well described in the results. Given this seems to be a central advance of the pipeline, it needs to be more clearly described in the results, with an accompanying schematic to more clearly show how it works. The benchmarking presented was also not as convincing as it could be, the datasets used are quite limited and other tools such as Kaiju and metaphlan would seem more appropriate comparators and it was somewhat surprising that there was no comparison to HUMAnN3 which is the current standard for processing metagenomic/metatranscriptomic datasets, albeit it also predicts function as well as taxonomy. Overall this is an interesting piece of software but the current manuscript does not seem to do it justice.

Specific comments:

1. The datasets used in the comparisons are quite limited. Simulated datasets especially can lead to false impressions of performance. Additional real datasets should be used, in addition it is essential to benchmark the tools against a real experimentally derived dataset, that features the complexity of a typical gut or soil microbiome. While it may not be possible to have gold standard assignments associated with these datasets, it would at least show what each tool is capable of.
2. The dataset used to illustrate a working use of AGMEMNON presented in supplemental is relatively trivial (UC v healthy), the authors should present a more meaningful dataset that exhibits less dramatic differences
3. The comparator tools should also include other taxonomic predictors such as Kaiju, three of the tools currently presented are iterations of similar algorithms.
4. It is considered essential that the authors compare their pipeline with HUMAnN3, as well as a more traditional process of metagenomics analysis where metagenomics datasets are first assembled and then a tool such as fastANI is then used to assign taxonomic labels to the various bins of assembled contigs, have the authors considered introducing an assembly step into their pipeline?
5. Figure 1 needs to be refocused on AGAMEMNON itself to illustrate what the algorithm is actually doing
6. The EM method needs to be more clearly explained in results
7. The shinyapp visualization for the results is interesting and it was surprising to see this denigrated to supplemental methods. I think this is an important part of the pipeline and one that is likely to appeal to users who might be more willing to adopt this tool if these features are

'advertised' more in the main manuscript (e.g. with figures showing various screenshots)

8. Documentation on the tool is quite limited and would be significantly improved by adding a user case scenario

9. The benchmarking assessments were quite limited and also non-intuitive, the mean square log error needs explaining better and in Figure 2, I had no idea what the two axes represent. Readers might also expect to see more traditional bar charts of taxa predicted and their relative abundances, and also, where there is a gold standard, some clearly representation of what % of which taxa were correctly predicted by each tool.

10. The results reporting strain (and even species) assignment accuracy were a little perplexing. This reviewer's expectation is that strains would be rarely predicted in real datasets as they are often not present in the reference databases. For the Illumina400 dataset, it was mentioned that 143 strains were missing from one of the reference datasets, thus it is not clear if the results in Figure 2 are accounting for these missing strains in some way. In any event, a more likely scenario is that the reference databases will lack strains found in real samples. Thus it would make more sense to take a generic database that does not contain the specific strains in the e.g. Illumina400 dataset, and see how well each method is able to predict at the species (or even genus) level. Presumably if you ran BWA against each of the reference genomes, you would end up with even better performance than AGAMEMNON, so the current comparisons seem unrealistic.

11. The emphasis on host filtering is a little misplaced as it is fairly trivial in most pipelines to add a host filtering step

12. In Figure 4 Gardnerella was reported in Peyers patch, how do the authors explain what is typically considered a microbe associated with the genital tract, to be found in the Ileum?

13. The single cell results were not well described, it was not clear what the read counting values represent and so it was not clear if this is a trivial exercise or not

Overview

We would like to thank the Reviewers for the thoughtful comments, feedback, and enthusiasm for our work. We have addressed all Reviewers' comments through the addition of numerous new data and analyses. The manuscript is accordingly revised and we believe that it has been improved substantially through the revision process. The manuscript now incorporates one of the largest collections of benchmarks conducted for a relevant algorithm, providing a holistic overview of the implementation's capabilities across different scenarios and use cases. It is also now more streamlined to install and deploy in different operating systems, while the git repository has been extended to incorporate code, binaries, docker instances, walkthroughs, and scripts to recreate all analyses mentioned in the manuscript. The key aspects of the revisions are firstly highlighted followed by the point-to-point response to the Reviewers' comments.

Key points in our revision include:

1. AGAMEMNON is now additionally provided in a Docker container, offering seamless and straightforward deployment across operating systems (including MacOS).
2. Expanded extensively the number of benchmarking tests as well as their scope and focus.
 - a. **Simulated tests:**
 - We incorporated an extensive index benchmark (NCBI RefSeq: 44,694 microbial sequences, > 8,500 microbial genomes). This test represents real-world scenarios where the index is agnostic to the application.
 - We performed additional tests where ~63% of the strains present in the Illumina 400 dataset were missing from the reference. At the same time, the reference is significantly larger compared to tests performed in the original submission (~5-fold increase compared to the previously largest index used in the original submission). This scenario showcases AGAMEMNON's ability to accurately detect the proper genus and species, despite the lack of the relevant strains in an extensive index.
 - b. **Tests using real microbe sequencing datasets:**
 - ***Synthetic Communities:*** we expanded the benchmarks using the synthetic community dataset against a significantly larger reference (complete NCBI RefSeq). AGAMEMNON results remain accurate in this demanding scenario.
 - ***Human microbiome samples:*** We incorporated in the benchmarks three experimentally derived shotgun metagenomics sequencing experiments originating from human stool samples to provide an overview of concordance between the different approaches.
 - c. **Included additional algorithms in the benchmarks:**
 - We extended the algorithms incorporated in the benchmarks to now represent multiple orthogonal families of methods: MetaPhlAn 3, a marker-based approach, Kaiju a protein-based sequence matching approach, Kraken 2 aiming for taxonomic assignment, and then Bracken and meta-Kallisto which aim to directly extract abundance estimates and are directly comparable to AGAMEMNON.
3. The Git repository has been revamped: all code has been made open source; additional binary files/docker containers have been included for easy deployment; all scripts for tests and benchmarks have been uploaded; a tutorial that walks users through the suite's feature set has been implemented.
4. The case studies were further expanded to include also the comparison of fecal metagenomic samples from patients with colonic adenomas, carcinomas, and healthy controls.

Please find our point-by-point response below, explaining all changes in further detail.

Reviewer 1

Reviewer #1: Skoufos et al. developed AGAMEMNON a method for metagenomic and metatranscriptomic quantification. Short reads are searched against a pufferfish index, which stores the reference genomes and queries them. Once seeds are found they are chained and scored by an alignment. Using these hits, a maximization expectation algorithm estimates the abundance of each species in the sample. Additionally, to the quantification method, the authors developed a visualization in R shiny.

Major:

- I tried to run the software on my MacOS notebook, however pre-compiled binaries are only provided for Linux. Please state what operating systems are supported. I could not find the required software versions of HISAT2, Pufferfish (and cedar?) to manually install so I could still try out the software. Please provide a list of versions of the software dependencies. Additionally, I would strongly recommend to package the software in bioconda to make it easy for users to install.

We thank the Reviewer for this suggestion. Since Linux is the most commonly used OS in servers, HPC, and lab environments, we had provided pre-compiled binaries just for this OS. Following the Reviewer's suggestion, we now added source code which could be used to compile and install in other OSs (including MacOS, which we tested), while to further increase ease of use, we created a Dockerized instance to enable easy deployment regardless of the user OS or setting. We tested the Docker instance across multiple OSs (including MacOS) to ascertain that the deployment of AGAMEMNON is straightforward in all instances.

- What is cedar (binaries/cedar)? Is that a novel part of this manuscript? If so, where is the source code?

Cedar is a novel algorithm specifically designed for metagenomic abundance estimation. Cedar is responsible for the abundance estimates right after the alignment step in the AGAMEMNON pipeline. It calculates posterior abundances as in the case of posterior counts for RNA-Seq data (efforts to solve this problem in gene expression space include RSEM and Salmon). However, since we're in the metagenomic space Cedar takes into consideration the lineage of the species/strain, while it expects the existence of unknown species/strains in a sample.

Even though Cedar is conceptually, functionally, and algorithmically an independent program from Pufferfish/Puffaligner, it is developed and maintained in a subdirectory of Pufferfish repository for two reasons. The first is technical, as both Cedar and Puffaligner share many dependencies, including libraries for parallel programming, file reading, etc. The second reason is the efficient binary communication between Cedar and Puffaligner that brings a performance boost for the AGAMEMNON pipeline. Even though Cedar accepts SAM files, it also accepts a binary file format (PAM) designed to transfer the minimum necessary alignment information between Puffaligner and Cedar. By reducing the size of the alignment file and only keeping the information required by Cedar as well as directly outputting the binary format, we reduce I/O use and analysis runtime. For the above, we decided that incorporating those in the same git section would be beneficial for the users and interested developers.

Pufferfish and Cedar binaries can be accessed by downloading release v1.0.1 from the pufferfish github repository (<https://github.com/COMBINE-lab/pufferfish>) and following the installation instructions in the README document. In addition to the Docker rules, we have also provided a bash script for the full process in the AGAMEMNON's github repository (<https://github.com/ivlachos/agememnon>).

- The authors mention that a biased reference can cause mis-quantification. However, in the benchmark only 1,840 reference genomes were used. This set is too small considering that GenBank contains hundreds of thousands of genomes. This reference set is also too clean since many draft genomes contain contamination, redundancy and are fragmented. From the current benchmark it is not clear to me how this method would perform on data like this.

Users who work with environmental microbial data (e.g., soil) may want to use GenBank as a reference. If the software does not perform well on this type of data, please declare it.

Please add a benchmark considering this attribute.

We thank the Reviewer for suggesting a more challenging and closer to a real-world application scenario. Following the Reviewer's suggestion, we substituted the small (1,800 genomes) optimal (100% of the tested genomes included) for a more realistic scenario. We now index the complete catalogue of NCBI RefSeq microbial genomes as a reference for both the simulated and the synthetic datasets. This extensive reference (reference 3, REF-3) comprises 44,694 sequences belonging to ~8,600 genomes. Results are reported in the manuscript for the algorithms Kraken 2, MetaPhlAn 3, Kaiju, Bracken and AGAMEMNON. Notably, we were not able to run meta-Kallisto, since it required more RAM than what was available in our largest server instance (i.e., > 512GB) when using this reference. This issue showcases the technical challenges and vast RAM requirements for alignment-based abundance quantification algorithms when using extensive indices, which are efficiently addressed in AGAMEMNON.

Importantly, we incorporated a case where 63% of the genomes in the simulated dataset and 100% of the genomes in the synthetic **are not present in the index**. These very challenging cases test the algorithms on whether they can quantify accurately the abundance on species and genus-level despite the lack of the relevant strain in the index, which is very common in real-world applications.

As shown in the figures below, even in these challenging settings, AGAMEMNON performed robustly across all performed tests. AGAMEMNON exhibited the smallest MSLE in all tests, in both the simulated and the synthetic experimental datasets. It had also the second lowest numbers of false positives (FP), behind MetaPhlAn 3. The low FP number for MetaPhlAn 3 is technically deflated, due to the pre-defined very small reference query space it uses compared to all other methods. MetaPhlAn 3 exhibits the lowest accuracy in all other metrics and tests performed.

As mentioned above, we were not able to include all methods for all tests. Meta-Kallisto could not index RefSeq with the RAM available (512GB). Kaiju was included only in the case of the synthetic dataset and not in the benchmark of the missing reference genomes, since its pre-set index includes all sequences of the simulated dataset (Illumina400) and cannot be excluded to render it comparable with the other methods. Finally, only AGAMEMNON and Kraken 2 can be used to extract strain-level abundances.

We believe that these challenging and much more realistic testing scenarios using simulated and experimental synthetic microbial communities show that AGAMEMNON can provide robust, reliable results.

Figure 1 (Fig 3 in revised manuscript): The Mean Squared Log Error (MSLE) and the number of false positive taxa (FP) between true and estimated read counts at the levels of Genus, Species and Strain using reference 3. We measured MSLE (a) using unfiltered results (0 x axis tick) and (b) by removing all instances where the true and estimated counts were both zero (1 x axis tick). False positive taxa were counted at all read thresholds between 0 and 10. At the read threshold of 0 reads (unfiltered results), all taxa were counted, even those with just 1 assigned read. At the read threshold of 1 read, we counted the taxa with > 1 assigned read and so on. Bracken, MetaPhlan 3 and Kaiju produce results up to the species level and thus they were not included in the strain-level comparisons.

In addition, we re-created all previous benchmark-relevant figures using the same metrics and filtering methodology and replaced the figures in the revised manuscript (shown also below). Similarly to the previous benchmarks, MetaPhlan 3 had the smallest number of false positives in all tested scenarios with AGAMEMNON following (Figures 2D-F, 3D-F). In the Illumina 400 dataset, AGAMEMNON displayed better performance in terms of MSLE in both genus and species resolution (Figures 2A, B). At the strain level, AGAMEMNON and Kallisto had the lowest MSLE (Figure 2C). In the synthetic dataset, all methods (excepting MetaPhlan 3) have similar MSLE values at the genus and species levels (Figures 3A, B). At the strain level, Kraken 2 had the best performance with Kallisto following (Figure 4C). It is worth noting that in this particular case (i.e., strain-level MSLE, synthetic dataset), the difference between methods is extremely small (all methods have an MSLE in the range of 4 – 5.8 at the 1 read threshold).

Illumina 400 dataset – Reference 1

Figure 2 (Fig. 2 in revised manuscript): The Mean Squared Log Error (MSLE) and the number of false positive taxa (FP) between true and estimated read counts at the levels of Genus, Species and Strain using the illumina 400 dataset and reference 1. We measured MSLE (a) using unfiltered results (0 x axis tick) and (b) by removing all instances where the true and estimated counts were both zero (1 x axis tick). False positive taxa were counted at all read thresholds between 0 and 10. At the read threshold of 0 reads (unfiltered results), all taxa were counted, even those with just 1 assigned read. At the read threshold of 1 read, we counted the taxa with > 1 assigned read and so on. Bracken and MetaPhlan 3 produce results up to the species level and thus they were not included in the strain-level comparisons.

Figure 3 (Supp fig. 1 in revised manuscript): The Mean Squared Log Error (MSLE) and the number of false positive taxa (FP) between true and estimated read counts at the levels of Genus, Species and Strain using the synthetic dataset and reference 2. We measured MSLE (a) using unfiltered results (0 x axis tick) and (b) by removing all instances where the true and estimated counts were both zero (1 x axis tick). False positive taxa were counted at all read thresholds between 0 and 10. At the read threshold of 0 reads (unfiltered results), all taxa were counted, even those with just 1 assigned read. At the read threshold of 1 read, we counted the taxa with > 1 assigned read and so on. Bracken and Metaphlan 3 produce results up to the species level and thus they were not included in the strain-level comparisons.

These benchmarks capture a wide range of less and more challenging scenarios and clearly show that even though AGAMEMNON is not just a method for the quantification of microbial abundances but a full-fledged analysis suite, its abundance estimation engine is robust and accurate, exhibiting the top performance in most tests, even when compared against implementations from different families of metagenomic algorithms.

- In figure 2 and 3 it is not intuitive that the MSLE increases with higher read threshold while false positive number decreases. How are unclassified queries labelled? It would be better to separate how many queries are classified and how accurate they are classified. Please explain in detail how the MSLE was computed.

The Reviewer's observation is correct and MSLE can indeed increase with higher read thresholds, since the total number of genomes in the query (denominator) is decreasing following filtering. For clarity, we present a detailed example below:

In the genus-level results, if we obtain for instance a squared log error of 800 for a total number of 400 genera, without filtering (zero threshold), to calculate the MSLE we would divide the squared log error by the total number of genera (MSLE = 2).

If we increase the read threshold from 0 to 5, a large number of genera are removed. This filtering not only decreases the squared log error but also the denominator (total number of identified genera). Since usually in metagenomic samples there is a long tail of lowly abundant species/strains, the reduction of the denominator is usually larger than the squared log error. In the same example above, for a threshold of 5, the squared log error is 650 for 220 genera, resulting in an MSLE of 2.95, which is ~50% higher.

For clarity, we created a separate section in the Methods (“Accuracy metrics”), explaining the metrics in greater detail and also incorporated in the figures only thresholds that would make sense to use in actual settings (e.g. up to 10 reads).

Minor:

- Please open-source the scripts to reproduce the results

All the scripts for all analyses have been added to the <https://github.com/gskoufos/AGAMEMNON-manuscript> github repository.

- Table 1 states that the benchmark set "illumina 400" has ~20M reads. However, on page 6 line 33 it is mentioned that it consists of 53.3 million paired-end reads. Does the QC of the reads remove 23.3 million reads? If yes, why is the QC that strict?

We would like to thank the reviewer for this remark. The correct number is 20M paired-end reads in both instances and the text has been corrected.

- Kraken2 seems to be consistently 3-4 times faster than AGAMEMNON, except in the rep-ref-ncbi-mock benchmark. Why is AGAMEMNON so much faster here? The size of the database seems to be about equal.

We would like to thank the reviewer for catching this typo. The text has been corrected.

- Adding a link to the software in the abstract would make it easier to find the software

A relevant link has been added in the abstract to enable readers to easily access the software.

- line 52 p 12. "Encode samples" should be "ENCODE samples"

The text has been edited accordingly.

- The name of x-axis should consistent between figure 2 and figure 3

The x axis has been edited to be consistent in the two figures.

- In figure 4, "The six most abundant species" is written in the legend. However, there are more than six species in the graph

The plot included the 6 most abundant species per tissue (which results to more than 6 species total). The legend has been edited for clarity.

- Please consider uploading all code in the git repository and not only providing it as part of the GitHub Released versions.

All code has been incorporated in the git as recommended. We agree that this will make it more accessible to interested users.

Reviewer 2

Reviewer #2: This paper describes AGAMEMNON, a software pipeline that predicts the relative abundance of taxa in metagenomic or metatranscriptomic samples. In microbiome research we are seeing more of a shift away from correlative studies to more causative studies, driven by the application of metagenomics and/or metatranscriptomics. Currently there is a need for pipelines that are able to process these datasets. While most interest focuses on functions captured by these datasets, there is still interest in assigning taxonomic labels to the sequences. At present there are a number of tools that seek to assign taxonomic labels to these datasets, including some that the authors have benchmarked their tool against here. There are also pipelines that take the user through the various sequence processing steps - notably HUMAnN2 (now HUMAnN3) which appears to do a similar analysis as AGAMEMNON (albeit it also predicts microbiome function). The main advance as far as this reviewer understands, is the application of an expectation maximization algorithm to predict the taxa present in a sample. While algorithms such as KRAKEN may assign many thousands of taxonomic labels to a dataset, AGAMEMNON takes the approach that many of those labels may be false positives and instead attempts to assign sequences to a more limited set of 'likely' taxa, which seems a similar idea to MetaPhlan, although through a different method. That said, the paper is presented in a way that makes it difficult for the reader to understand what the authors are attempting to present. On the one hand there is a new algorithm for assigning taxonomic labels, on the other there is a sequence processing pipeline. Both aspects seem to have got obscured in the writing, such that it is not clear what the authors are presenting.

I would recommend that the authors consider rewriting the manuscript, that more clearly describes AGAMEMNON as a software pipeline for processing metagenomic/metatranscriptomic datasets. The EM algorithm is also buried in the methods and not well described in the results. Given this seems to be a central advance of the pipeline, it needs to be more clearly described in the results, with an accompanying schematic to more clearly show how it works. The benchmarking presented was also not as convincing as it could be, the datasets used are quite limited and other tools such as Kaiju and metaphlan would seem more appropriate comparators and it was somewhat surprising that there was no comparison to HUMAnN3 which is the current standard for processing metagenomic/metatranscriptomic datasets, albeit it also predicts function as well as taxonomy. Overall, this is an interesting piece of software but the current manuscript does not seem to do it justice.

By following the reviewer's apt remarks, we have restructured the manuscript and brought also the EM algorithm to the spotlight. We also expanded the text clarifying the differences of our approach to existing tools, since AGAMEMNON is not an algorithm for taxonomic rank assignment (such as Kraken) but aims for abundance estimation as in the case of Bracken or Meta-Kallisto. For our initial submission, we had tested also MetaPhlAn v1 and v2, but we decided not to include the results, since they are based on direct alignment against a marker gene set collection and in most of the tested cases performed quite poorly against all other algorithms in all comparisons.

We agree with the reviewer that all these tools are utilized by researchers to estimate relative or absolute abundances in their metagenomic data, even if their development aims are different (taxonomic rank assignment, abundance estimation, etc.). To this end, we have extended our benchmarks with MetaPhlAn v3 and Kaiju and expanded the explanation of the differences in each approach and the optimal settings for their use. In all settings and in most of the diverse benchmarks, AGAMEMNON is a top performer, showing that despite being an

extensive analysis suite, its abundance estimation engine is accurate in genus, species and strain resolution, while being fast and frugal in required resources. We believe that these comments and the relevant additions greatly improved the readability of the manuscript and clearly demonstrate the breakthroughs in performance, accuracy and scope, achieved by AGAMEMNON.

Specific comments:

1. The datasets used in the comparisons are quite limited. Simulated datasets especially can lead to false impressions of performance. Additional real datasets should be used, in addition it is essential to benchmark the tools against a real experimentally derived dataset, that features the complexity of a typical gut or soil microbiome. While it may not be possible to have gold standard assignments associated with these datasets, it would at least show what each tool is capable of.

We agree with the reviewer that when presenting an algorithm, the scope of the performed tests is key. To this end, we added to the comparisons three experimentally derived datasets (Figures 5 and 6 below) but also used the simulated and synthetic datasets with additional references and settings. Synthetic communities are a useful tool in this setting, since they are not simulated data (i.e., they are actual samples going through the complete experimental process) with known species/strains at predefined abundances. Certainly, they are less complex than a soil or gut sample, but they mirror the challenges of actual biological samples, including technical noise from DNA/RNA extraction and library preparation, while incorporating numerous known and unknown biases. Importantly, they represent a fair test set across all settings, since the generation of simulated data could potentially favor algorithms based on similar principles. Finally, we also utilize for the first time a single microbe sequencing experiment (SIC-Seq), where the abundances estimated by AGAMEMNON were compared against the quantification performed by the creators of the method using bright light microscopy.

In summary, throughout the whole manuscript, we compare, test and use AGAMEMNON utilizing the following datasets:

- a) The Illumina 400 dataset (simulated) using different references (from targeted to complete NCBI RefSeq) and added cases where strains are missing from the reference.
- b) The synthetic dataset (experimentally-derived dataset with known species/abundances) analysed using diverse references.
- c) Two Host-microbiome mixed datasets (simulated) with low (3.77%) and high microbial read abundance (7.53%).
- d) The SIC-Seq dataset (single-microbe sequencing experimentally derived dataset with known species/abundances)
- e) Human fecal sample SRS011084 (experimentally-derived dataset from the Human Microbiome Project)
- f) Human fecal sample SRS013639 (experimentally-derived dataset from the Human Microbiome Project)
- g) Human fecal sample SRS147139 (experimentally-derived dataset from the Human Microbiome Project)
- h) 47 ulcerative colitis and healthy control fecal samples from the Inflammatory Bowel Disease Multiomics Database
- i) 24 colon carcinoma, adenoma and healthy control fecal samples from Feng *et al.* (Nat. Comms, 2015)

To our knowledge it is the most comprehensive testing/benchmarking effort for a relevant implementation.

The newly incorporated human fecal samples were an especially useful addition, since they enabled us to evaluate concordance between methods, even if the true abundances are not known. As shown in Figure 5 below, all methods (excepting MetaPhlAn 3) exhibit a positive Spearman's rho > 0.5 in almost all samples and at both taxonomic ranks. As expected, the highest correlation is between Kraken 2 and Bracken, since Bracken abundances are based on Kraken 2 taxonomic ranks. AGAMEMNON exhibits a strong positive correlation with both Bracken and Kraken 2 at both the genus (> 0.7) and species (> 0.5) levels. These results demonstrate that most of the methods have a relative agreement in the three experimentally-derived human datasets.

Figure 5 (Fig. 4 in revised manuscript): The pairwise Spearman correlation of each method in three human fecal samples at the levels of genus and species. Before calculating Spearman's rho, we removed all instances of zero-abundant taxa from all methods.

Surprisingly, as shown in Figure 6, all five methods share a relatively small number of commonly identified genera (< 50) and species (< 60). This is due to MetaPhlAn v3 which identifies a significantly smaller number of taxa. When removing MetaPhlAn from the overlap calculations, we see that Kaiju, Kraken 2, Bracken and AGAMEMNON exhibit high concordance, sharing $> 1,000$ and $> 2,800$ commonly reported genera and species respectively. Kaiju reported the largest number of distinctly identified species (> 530). Even though these are real datasets for which the ground truth is unknown, the numbers reported by Bracken, Kraken 2, Kaiju and AGAMEMNON are closer to what the human microbiome community considers a standard complexity and diversity of a typical human gut microbiome.

Figure 6 (Supp. fig 2 in revised manuscript): Venn diagrams demonstrating the commonly & distinctly identified taxa between Metaphlan 3, Bracken, Kraken 2, Kaiju and AGAMEMNON at the levels of genus and species.

2. *The dataset used to illustrate a working use of AGMEMNON presented in supplemental is relatively trivial (UC v healthy), the authors should present a more meaningful dataset that exhibits less dramatic differences*

We appreciate the Reviewer's comment. We have expanded the test cases by adding a second independent use case using 24 fecal samples originating from colon adenoma and carcinoma patients, as well as healthy controls. Details regarding the samples, the conducted analysis, and the results of the case study are presented in detail in the **Supplementary document 2**.

3. *The comparator tools should also include other taxonomic predictors such as Kaiju, three of the tools currently presented are iterations of similar algorithms.*

Kaiju and MetaPhlan v3 have been added in the benchmarking following the Reviewer's suggestion. We agree with the reviewer that Kraken, Kraken 2, and Bracken are algorithmically similar and, thus, we removed Kraken from the benchmarks and kept Kraken 2 and Bracken, since Kraken 2 aims for taxonomic ranks and Bracken is an algorithm for metagenomic abundance estimation. We also refer the reviewer to our detailed response to Reviewer 1 (Comment #3), which shows the newly added benchmark results and the original figures now expanded with additional metrics and algorithms.

4. *It is considered essential that the authors compare their pipeline with HUMAnN3, as well as a more traditional process of metagenomics analysis where metagenomics datasets are first assembled and then a tool such as fastANI is then used to assign taxonomic labels to the various bins of assembled contigs, have the authors considered introducing an assembly step into their pipeline?*

By following the Reviewer's apt remarks, we have added to the comparisons MetaPhlan which HUMAnN3 utilizes for the microbial abundance estimation step (HUMAnN3 aims to profile microbial metabolic pathways

and functions based on the MetaPhlAn abundances). We incorporated the newest version of MetaPhlAn (v3) in all tests. MetaPhlAn exhibits the lowest accuracy across all benchmarks, apart from false positives and exhibits the smallest overlap with all other tested methods.

AGAMEMNON not only comes with a pre-compiled index (as all tested methods) but can also use any *ad hoc* index prepared as a FASTA file, which many of the tested methods do not support (e.g. MetaPhlAn v3 and Kaiju). Even though we consider that creating an assembly method or incorporating one in the AGAMEMNON pipeline is beyond its scope, AGAMEMNON already supports the use of *ad hoc* indices, so users could produce assemblies with their favorite tools and utilize AGAMEMNON to quantify abundances.

5. Figure 1 needs to be refocused on AGAMEMNON itself to illustrate what the algorithm is actually doing

Figure 1 of the manuscript has been completely redesigned to show also the inner workings of the algorithm and its quantification engine, while retaining the high-level overview of the modules and functionalities. The reviewer can find the figure below:

Figure 7 (Fig. 1 in revised manuscript): Schematic representation of AGAMEMNON. Dataset input is in raw FASTQ format. Paired-end (PE) or Single-end (SE) libraries are supported. For single cell libraries, AGAMEMNON has helper scripts to enable per-cell analyses. In case of host tissue samples or contaminant quantification activities, the reads are first aligned against the host genome and the contaminant reference index using HISAT2. The host alignment file is saved for downstream applications and the resulting unmapped reads are forwarded to the main metagenomics/metatranscriptomics pipeline. Selective alignment is performed on the microbial reads against the reference index, while microbial abundances are subsequently quantified. A raw quantification table is produced as well as a taxonomic rank table. The results of the analysis can be used as input to AGAMEMNON's R-Shiny application, which enables diverse analyses and investigations from a graphic user interface, including visualizations, dimensionality reduction, differential abundance and diversity index analyses.

6. *The EM method needs to be more clearly explained in results*

We have expanded the relevant sections and increased the number of mentions and level of detail across the manuscript. We also added the following paragraph to the Results section:

AGAMEMNON's quantification engine

AGAMEMNON uses an expectation maximization (EM) algorithm to probabilistically resolve the origin of reads (Methods, "Quantification of microbial fragments") to individual references. This step contributes to its enhanced quantification accuracy at the species and strain levels. Unlike methods such as Kraken and Kraken 2 that propagate reads that have multiple best assignments to some higher taxonomic rank, AGAMEMNON, via the inference performed in Cedar, makes use of other reads and their allocations to determine the probability that the ambiguous read arises from a specific reference. Similar to other EM algorithms, in Cedar we iterate over these two steps of Expectation and Maximization until the convergence criteria are met. In each iteration, we calculate the read probability distribution in the Expectation step and distribute reads across strains to maximize the probability of observed reads in the Maximization step. Users can customize whether this procedure can be run at different levels of the taxonomy tree.

Cedar's EM procedure is specifically modified according to fundamental properties and challenges of metagenomic quantification. For instance, in metagenomic indexes there is often high similarity among the strain sequences belonging to the same species (sometimes even across species), which increases the complexity of disentangling reads at lower levels of the taxonomy. Additionally, reads coming from unknown species or unknown strains can be falsely assigned to entries existing in the index, resulting in false positive non-zero values. As part of Cedar's pipeline, we address these challenges through iterative, mass-preserving filtering. We look for groups of references that share the same class of reads and are fully ambiguous without any preference towards a reference over the others to detangle reads in Maximization step. We adopt the "set-cover" algorithm to find the minimum number of references that explains the reads in this group. Through this approach, we tackle the problem by sparsifying the solution (i.e., the set of species that may be assigned a non-zero abundance) in a manner that still retains all of the mapped reads.

7. *The shinyapp visualization for the results is interesting and it was surprising to see this denigrated to supplemental methods. I think this is an important part of the pipeline and one that is likely to appeal to users who might be more willing to adopt this tool if these features are 'advertised' more in the main manuscript (e.g. with figures showing various screenshots)*

We agree with the reviewer that it is indeed a very useful aspect of the analysis suite. We have brought the relevant section into the main text, as well as added Figure 6 (also added below) which shows screenshots of analyses and use-cases. Importantly, we have added mentions within the text and the discussion.

Figure 8 (Fig. 6 in revised manuscript): Screenshots of AGAGEMNON's Shiny application. (Top row) Visualization of microbial abundances through the use of Manhattan plots and Boxplots. (Middle row) Heatmap visualization and clustering using top N (in terms of abundance) microbes and PCA/MDS analysis. (Bottom row) Diversity index analysis and Interactive tables showing the full lineage of microbes identified in the analysed samples and Differential expression analysis module and results.

8. Documentation on the tool is quite limited and would be significantly improved by adding a user case scenario

We have expanded the documentation in the git with a step-by-step use-case in high detail. The use case scenario can be accessed from the Wiki page of AGAGEMNON's github repository: <https://github.com/ivlachos/agememnon/wiki>

9. *The benchmarking assessments were quite limited and also non-intuitive, the mean square log error needs explaining better and in Figure 2, I had no idea what the two axes represent. Readers might also expect to see more traditional bar charts of taxa predicted and their relative abundances, and also, where there is a gold standard, some clearly representation of what % of which taxa were correctly predicted by each tool.*

The explanation of the MSLE has been significantly expanded and we have ascertained that both axes in all figures are clearly mentioned in the figure legends. We also added additional accuracy metrics (false positives and overlap between methods) and revised the previously incorporated benchmarks. For clarity, we also created a separate section in the Methods (“Accuracy metrics”), explaining the metrics in greater detail. Please also see to our responses to Reviewer 1 (answers #3 and #4).

10. *The results reporting strain (and even species) assignment accuracy were a little perplexing. This reviewer's expectation is that strains would be rarely predicted in real datasets as they are often not present in the reference databases. For the Illumina400 dataset, it was mentioned that 143 strains were missing from one of the reference datasets, thus it is not clear if the results in Figure 2 are accounting for these missing strains in some way. In any event, a more likely scenario is that the reference databases will lack strains found in real samples. Thus it would make more sense to take a generic database that does not contain the specific strains in the e.g. Illumina400 dataset, and see how well each method is able to predict at the species (or even genus) level, Presumably if you ran BWA against each of the reference genomes, you would end up with even better performance than AGAMEMNON, so the current comparisons seem unrealistic.*

By following the reviewer's suggestion, we created a case where ~63% of the strain sequences in the Illumina 400 dataset were missing from the reference database (reference 3). In addition, 100% of the strains and even species of the synthetic dataset were missing from the same reference. In both these cases, AGAMEMNON results remain accurate at genus and species resolution. We agree with the reviewer that the strains that are actually present in experimentally-derived datasets are rarely present in the reference. This is especially true for environmental samples. In the case of biomedical microbiome research though, (i.e., human-, mouse-derived samples etc.) we believe that the efforts of the community especially during the past few years (Almeida *et al.* Nature Biotechnology, 2020, Zhu *et al.* bioRxiv, 2019, Pasolli *et al.* Cell, 2019, etc.) have tremendously expanded our characterization of the human microbiome and could soon support species- or even strain-level references specifically tailored for human and mouse microbiome research. We also refer the Reviewer to our detailed response to reviewer 1 (comment #3).

Regarding BWA alignment per genome, reads could map on multiple genomes (sequences are extensively shared across strains and species) and the final number of high quality mappings will be vastly larger than the number of original reads. As we explain in the text above for the E/M step, algorithms such as Kraken address this problem by propagating the assignments higher to the taxonomic tree, losing resolution at the lower branches (i.e. strain), while AGAMEMNON with the E/M step calculates the posterior probability for a read to be assigned based on mappings of additional reads and by leveraging additional information. This approach enables AGAMEMNON to go beyond taxonomic ranks or marker-based assignment and provide accurate abundances down to strain resolution.

11. *The emphasis on host filtering is a little misplaced as it is fairly trivial in most pipelines to add a host filtering step*

We agree with the reviewer that adding some sort of host filtering is relatively trivial and there are multiple ways this can be achieved. However, this is the first time a host tissue/biofluid approach has been thoroughly tested and shown to perform with such high accuracy. Comparing against GATK-PathSeq, a tool specifically designed to perform only this analysis (microbe abundance detection from host samples) created from a very experienced group (the first version of PathSeq was published in Nature Biotechnology in 2011 and the second in Bioinformatics in 2018), AGAMEMNON exhibits dramatically improved accuracy across all tested metrics.

Therefore, for specialists and non-specialists alike, we believe that having a tested and highly optimized (not only for accuracy but also for time, CPU, RAM and I/O) implementation can be transformative, especially when outperforming leading implementations specialized in such analyses.

12. In Figure 4 Gardnerella was reported in Peyers patch, how do the authors explain what is typically considered a microbe associated with the genital tract, to be found in the Ileum?

There are multiple mentions in the recent literature of Gardnerella in the GI tract. For instance, in Zhou *et al.* (Front Nutr 2021), the authors identified Gardnerella as a characteristic GI microbe for ulcerative colitis patients, while in Gopalakrishnan *et al.* (Science, 2018) Gardnerella was also identified in the gut microbiome of cancer patients. Figure 4 of the original manuscript has been moved to supplementary material (Supp. Fig 4)

13. The single cell results were not well described, it was not clear what the read counting values represent and so it was not clear if this is a trivial exercise or not

We appreciate the reviewer's comment. Lan *et al.*, calculated microbial abundances directly from bright light microscopy (as well as other indirect methods) and the relevant metrics were derived from the original publication. We provide a detailed description in the Methods section, named "Single-cell artificial microbial community analyses" (pasted below). We added also a mention in the text and the relevant figure, in order to further assist the Readers. The relevant text reads:

"We used AGAMEMNON to analyze an artificial microbial community originating from a microbial single-cell sequencing technique (SiC-Seq). The community consists of 8 bacterial species and 2 yeasts. We created a reference comprising 116 complete genomes (278 sequences) including the genomes used to create the synthetic community. The rest of the genomes were microorganisms belonging to the same genera as the species used for the construction of the synthetic microbial community (Supplementary Table "single-cell genomes"). We quality-checked and pre-processed the SiC-Seq dataset by grouping the reads under the barcode they belong and removed all barcodes with <50 reads, yielding a little more than 48,000 unique barcode groups (cells). We then applied AGAMEMNON to identify and quantify the abundances of the microbial species. Finally, similarly to the analyses conducted in Lan et al. we considered the most commonly mapped species (in terms of read counts) as the species from which the cell originates. We then counted the number of cells per species, divided with the total number of cells and multiplied by 100 to acquire the species' relative abundances. We utilized the bright-field microscopy results from SiC-Seq paper as the reference annotation for this dataset and to derive AGAMENON's accuracy in single-cell data."

Second round of review

Reviewer 1

I would like to thank the authors for answering all my questions.
There is only one minor point remaining.

Minor:

- In supplemental table "Index benchmark", "REF-3" Kraken2 is highlighted as fastest tool but Kaiju is faster

Reviewer 2

In this resubmission, the authors have addressed several of the concerns that were raised in the original round of reviews. However, there are still some outstanding concerns that have only been partially addressed:

1) The datasets used in the comparisons are quite limited.

The authors have partially addressed this concern with the inclusion of two new figures that focus on concordance of results. It is noted however that three tools (Kaiju, Bracken and Kracken) share a significant number of assignments that agamemnon does not. How do the authors explain this? At the same time, the authors do not appear to have included any experimentally derived dataset for which taxon abundance has been measured independently - only the synthetic dataset is presented. It is suggested that the authors search for some type of gnotobiotic dataset (e.g. based on germ free mice in which known microbes have been used to seed the developing GI tract). The authors could then compare e.g. BWA, agamemnon and other tools with and without the known microbe genomes in their reference databases.

2) The dataset used to illustrate a working use of AGMEMNON presented in supplemental is relatively trivial

This has been adequately addressed

3) The comparator tools should also include other taxonomic predictors such as Kaiju, three of the tools currently presented are iterations of similar algorithms

This has been addressed for the most part, but note comments below

4) It is considered essential that the authors compare their pipeline with HUMAnN3

The host RNA/DNA section appears to describe an attempt to separate bacterial from host reads. agamemnon is compared to a GATK Pathseq pipeline. It is not clear why the Pathseq pipeline was selected. What was the reason for not including HUMAnN3 (i.e. using the associated kneaddata pipeline) in these analyses?

5) Figure 1 needs to be refocused on AGAMEMNON itself to illustrate what the algorithm is actually doing

Figure 1 is much improved, but the schematic on EM could be improved to provide more detail on what is actually happening under the hood

6) The EM method needs to be more clearly explained in results

The attempts to clarify are appreciated. However, there is still not enough clarity for the average user (and general audience of Genome Biology) of the tool to understand the implementation of the EM algorithm. The current explanation appears relatively brief and needs to be revisited and passed by some colleagues in neighboring labs to ensure that it is clear how the algorithm is working. More detail in schematic 1 would clearly help.

7) The shinyapp visualization for the results is interesting and it was surprising to see this denigrated to supplemental methods.

The figure provided was great! and will be helpful in getting new users to adopt the tool

8) Documentation on the tool is quite limited

This has been adequately addressed

9) The benchmarking assessments were quite limited and also non-intuitive, the mean square log error needs explaining better and in Figure 2, I had no idea what the two axes represent. Readers might also expect to see more traditional bar charts of taxa predicted and their relative abundances, and also, where there is a gold standard, some clearly representation of what % of which taxa were correctly predicted by each tool.

The explanation of MLSE needs to be added to the Results section. I also agree with Reviewer 1 that benchmarking would benefit from additional measures of performance. As suggested previously it would be helpful for the more general readers of Genome Biology to be presented with the ore traditional bar charts of taxon predictions along with some graphic showing the accuracy of assignments. As the response to both reviewers to questions over the MLSE shows, its use is relatively abstract and likely to confuse the readers who would benefit from the addition of more straightforward graphical analyses.

10) The results reporting strain (and even species) assignment accuracy were a little perplexing. For the Illumina400 dataset it is not clear what the authors mean when they state they removed 63% of the strain sequences, ideally it would be the entire genome (or complete set of sequences for a specific strain) that were removed, but it seems that instead the authors randomly removed ~2/3 of each of the 400 strains sequences present in the dataset - is this correct? If the entire genome was removed, then it is not clear how Figure 2 can provide metrics for the performance of the tools at strain level (since all the reference strain sequences should have been removed) - maybe I am missing something here, but as presented this is not reflecting what happens when you don't have the strain you are searching for in your reference database.

Regarding the authors response: "Regarding BWA alignment per genome, reads could map on multiple genomes (sequences are extensively shared across strains and species) and the final number of high quality mappings will be vastly larger than the number of original reads. " This would actually be good to demonstrate in your benchmarking

11. The emphasis on host filtering is a little misplaced

See comment 4 above

12. In Figure 4 Gardnerella was reported in Peyers patch

The references cited were not convincing, largely reliant on 16S rDNA surveys, although the Front Nutr study did suggest that they confirmed with qPCR, although that may have been confounded by the choice of primers used. Given lack of wider supporting literature, these may therefore represent mis-annotations. It would be important to provide some sort of note explaining this finding, with these caveats, in the authors own data.

13. The single cell results were not well described

This has been adequately addressed

Response to Reviewer 1

Reviewer #1: I would like to thank the authors for answering all my questions. There is only one minor point remaining.

We are extremely pleased that the Reviewer finds that the revised manuscript addresses all provided comments.

Minor:

- In supplemental table "Index benchmark", "REF-3" Kraken2 is highlighted as fastest tool but Kaiju is faster

We have updated the relevant table and highlighted Kaiju in that particular scenario.

Response to Reviewer 2

Reviewer #2: In this resubmission, the authors have addressed several of the concerns that were raised in the original round of reviews. However, there are still some outstanding concerns that have only been partially addressed:

We would like to thank the Reviewer for evaluating our revised manuscript.

1) The datasets used in the comparisons are quite limited.

The authors have partially addressed this concern with the inclusion of two new figures that focus on concordance of results. It is noted however that three tools (Kaiju, Bracken and Kraken) share a significant number of assignments that agamemnon does not. How do the authors explain this?

We appreciate the reviewer's comment. Indeed, Kaiju, Bracken and Kraken 2 share on average ~300 taxa at the genus level and ~2,300 taxa at the species level that AGAMEMNON does not. Since these are real human gut samples with unknown abundances, there is no way to tell whether all these taxa are true or false positives. All four methods (AGAMEMNON, Bracken, Kraken 2 and Kaiju) share on average ~1,100 and ~3,000 taxa at the levels of genus and species respectively. On average AGAMEMNON identified ~3,000 species while Kaiju, Bracken and Kraken 2 identified ~5,200.

It is often mentioned in the literature that the human gut comprises ~500 – 1,000 bacterial species (Gilbert et al., Nature medicine, 2018 and Turnbaugh PJ et al., Nature, 2007). Some recent studies suggest that this may be an underestimation of the actual human microbiome complexity but even so, the refined estimates, for the human gut, are much smaller compared to the ~5,200 species identified by the three methods (Stephen Nayfach *et al.*, nature, 2019 and Edoardo Pasolli *et al.*, Cell, 2019), pointing to an inflation by potentially thousands of false positives. Furthermore, our benchmarks show that Kraken 2, Bracken, and Kaiju consistently report a significantly higher number of false positives compared to AGAMEMNON. During our internal benchmarks when designing the algorithm, we identified the high number of false positives as one of the major issues with our first iteration of AGAMEMNON but also with all other available methods (excepting MetaPhlAn, where there the main issue is lack of sensitivity). This led to the design and implementation of the current algorithm prioritizing a parsimonious read assignment, leading to very high accuracy, as shown in the simulated and synthetic benchmarks.

Importantly, Bracken utilizes Kraken 2's taxonomic assignment results to render abundance estimates and thus, the results of these two methods are expected to be near identical especially in terms of presence/absence of microbial taxa. Based on the current biological knowledge and the benchmark results, we believe that a large part of the species reported uniquely by Bracken/Kraken 2 and/or Kaiju have an increased probability of being inflated with false positives.

Nonetheless, we cannot exclude the possibility that some of these species are actually present in the sample and we therefore added the following text in the manuscript:

“As expected, the highest correlation is between Kraken 2 and Bracken, since Bracken utilizes Kraken 2 output as the foundation of its abundance estimation calls. AGAMEMNON has a strong positive correlation with both Bracken and Kraken 2 at both the genus (> 0.7) and species (> 0.5) levels. These results demonstrate that most of the methods have a relative agreement in three experimentally derived human datasets. The very large number of species identified by Kaiju and Kraken 2/Bracken, could be the result of an inflation due to false positives, since they are significantly larger than our current expectations for the human microbiome (REFERENCE). However, since the ground truth is unknown it is not possible to identify the most accurate approach through this evaluation alone”

At the same time, the authors do not appear to have included any experimentally derived dataset for which taxon abundance has been measured independently - only the synthetic dataset is presented. It is suggested that the authors search for some type of gnotobiotic dataset (e.g., based on germ free mice in which known microbes have been used to seed the developing GI tract). The authors could then compare e.g., BWA, agamemnon and other tools with and without the known microbe genomes in their reference databases.

Our previous version of the manuscript comprised the synthetic communities and also the single microbe sequencing data (SiC-Seq) where the actual microbial abundances in both datasets are measured independently prior to sequencing by qPCR and bright light microscopy, respectively.

We appreciate the Reviewer’s suggestion for gnotobiotic mouse datasets, which we found highly relevant. We extensively searched GEO, ENA, and SRA, and identified the following two studies analyzing fecal samples from gnotobiotic mice:

Nathan P McNulty, Meng Wu, Alison R Erickson, Chongle Pan, Brian K Erickson, Eric C Martens, Nicholas A Pudlo, Brian D Muegge, Bernard Henrissat, Robert L Hettich, Jeffrey I Gordon, **Effects of diet on resource utilization by a model human gut microbiota containing *Bacteroides cellulosilyticus* WH2, a symbiont with an extensive glyco biome**, Plos Biol, 2013 (GEO entry: GSE48537)

Nathan P McNulty 1 , Tanya Yatsunenko, Ansel Hsiao, Jeremiah J Faith, Brian D Muegge, Andrew L Goodman, Bernard Henrissat, Raish Oozeer, Stéphanie Cools-Portier, Guillaume Gobert, Christian Chervaux, Dan Knights, Catherine A Lozupone, Rob Knight, Alexis E Duncan, James R Bain, Michael J Muehlbauer, Christopher B Newgard, Andrew C Heath, Jeffrey I Gordon, **The impact of a consortium of fermented milk strains on the gut microbiome of gnotobiotic mice and monozygotic twins**, Sci Transl Med, 2011 (GEO Entry: GSE31943)

Unfortunately, both studies utilized Illumina’s GAII and GAIIx, which produced 35bp-long reads (< 25 bp-long after pre-processing), too short to be analyzed with most of the benchmarked tools.

All other datasets we identified, such as (GSE48809), sequenced tissue and not fecal samples, which unfavorably affect all methods that do not have a host-specific function (all methods excepting AGAMEMNON and MetaPhlan/Humann3). Despite our efforts, we were not able to identify a dataset which could be included. However, since this is an important family of experiments, we added a relevant mention in the manuscript referring to the contribution of gnotobiotic mice and recent studies following this approach, which we believe will be informative to the Readers.

We also agree with the Reviewer that extensive benchmarking is always an important asset in methods manuscripts. During our extensive search of the aforementioned repositories, we identified 3 additional experimental datasets which could be used to further expand our testing:

1. Two shotgun metagenomics samples (Marcus B. Jones et al., PNAS, 2015) conducted on a bacterial community composed of a mixture of 20 microbial species that vary in genome size and abundance.
2. Four shotgun metagenomics samples (Wenyi Xu *et al.*, frontiers in Microbiology, 2021) conducted on two different microbial communities. The first community comprises 69 human gut bacterial species belonging to 33 genera with equal abundances (1.45% each taxon). The second community comprised 62

human gut bacterial species belonging to 28 genera with varied abundances (0.0004% - 20.35%). To the best of our knowledge and based on Wenyi Xu *et al.*, these are the most complex and realistic mock bacterial communities with known abundances ever constructed.

3. One shotgun metagenomics sample (Aaron M. Walsh *et al.*, Microbiome, 2018) generated by mixing equimolar ratios of genomic DNA from 13 food-related bacteria.

In all seven samples above, bacterial abundance measurements were conducted independently prior to sequencing.

Using these newly incorporated samples, we benchmarked AGAMEMNON against Kraken 2, Bracken, MetaPhlAn 3, and Kaiju. Since the actual bacterial abundance levels in all 7 samples were measured at the species level, the reported comparison results are for the levels of species and genus. As previously, we used the complete catalogue of NCBI RefSeq microbial genomes as a reference (reference 3, REF-3) which comprises 44,694 sequences belonging to ~8,600 genomes. In addition, we created a subset of REF-3 by removing 9 bacterial species (all genomes and/or complete set of sequences belonging to these species) that were present in the bacterial community from Marcus B. Jones *et al.*, PNAS, 2015 to mimic the realistic scenario of bacterial species present in the samples but absent from the reference. The newly created reference (reference 4, REF-4) comprises 38,691 sequences. We were not able to run meta-Kallisto, since it required more RAM than what was available in our largest server instance (i.e., > 512GB) when using these references.

As shown in the figures below, even in these challenging settings, AGAMEMNON performed robustly across all performed tests. Even though AGAMEMNON offers numerous functionalities and not just quantification results, at the genus level, it exhibited the smallest MSLE in all of the tests and all experimentally derived datasets. At the species level, AGAMEMNON exhibited better or almost-equally good in most derived statistics and scenarios.

These additional tests further show that AGAMEMNON not only is highly accurate but effectively tackles the high number of false positives observed in such analyses. It presented the second lowest numbers of false positives (FP), behind MetaPhlAn 3 in all taxonomic ranks and tested datasets. The low FP number for MetaPhlAn 3 is technically deflated, due to the pre-defined very small reference query space it uses compared to all other methods. MetaPhlAn 3 exhibits the lowest accuracy in all other metrics and tests performed.

Figure 1 (Supp. material fig. 3 in revised manuscript): The Mean Squared Log Error (MSLE) and the number of false positive taxa (FP) between true and estimated read counts at the levels of Genus and Species using **reference 3** and samples from Marcus B. Jones *et al.*, PNAS, 2015. We measured MSLE (a) using unfiltered results (0 x axis tick) and (b) by removing all instances where the true and estimated counts were both zero (1 x axis tick). False positive taxa were counted at all read thresholds between 0 and 10. At the read threshold of 0 reads (unfiltered results), all taxa were counted, even those with just 1 assigned read. At the read threshold of 1 read, we counted the taxa with > 1 assigned read and so on.

Figure 2 (Supp. material fig. 5 in revised manuscript): The Mean Squared Log Error (MSLE) and the number of false positive taxa (FP) between true and estimated read counts at the levels of Genus and Species using **reference 3** and samples from Wenyi Xu *et al.*, frontiers in Microbiology, 2021. We measured MSLE (a) using unfiltered results (0 x axis tick)

and (b) by removing all instances where the true and estimated counts were both zero (1 x axis tick). False positive taxa were counted at all read thresholds between 0 and 10. At the read threshold of 0 reads (unfiltered results), all taxa were counted, even those with just 1 assigned read. At the read threshold of 1 read, we counted the taxa with > 1 assigned read and so on.

Figure 3 (Supp. material fig. 6 in revised manuscript): The Mean Squared Log Error (MSLE) and the number of false positive taxa (FP) between true and estimated read counts at the levels of Genus and Species using **reference 3** and samples from Wenyi Xu *et al.*, *frontiers in Microbiology*, 2021. We measured MSLE (a) using unfiltered results (0 x axis tick) and (b) by removing all instances where the true and estimated counts were both zero (1 x axis tick). False positive taxa were counted at all read thresholds between 0 and 10. At the read threshold of 0 reads (unfiltered results), all taxa were counted, even those with just 1 assigned read. At the read threshold of 1 read, we counted the taxa with > 1 assigned read and so on.

Figure 4 (Supp. material fig. 7 in revised manuscript): The Mean Squared Log Error (MSLE) and the number of false positive taxa (FP) between true and estimated read counts at the levels of Genus and Species using **reference 3** and the sample from Aaron M. Walsh *et al.*, Microbiome, 2018. We measured MSLE (a) using unfiltered results (0 x axis tick) and (b) by removing all instances where the true and estimated counts were both zero (1 x axis tick). False positive taxa were counted at all read thresholds between 0 and 10. At the read threshold of 0 reads (unfiltered results), all taxa were counted, even those with just 1 assigned read. At the read threshold of 1 read, we counted the taxa with > 1 assigned read and so on.

Figure 5 (Supp. material fig. 4 in revised manuscript): The Mean Squared Log Error (MSLE) and the number of false positive taxa (FP) between true and estimated read counts at the levels of Genus and Species using **reference 4** and

samples from Marcus B. Jones et al., PNAS, 2015. We measured MSLE (a) using unfiltered results (0 x axis tick) and (b) by removing all instances where the true and estimated counts were both zero (1 x axis tick). False positive taxa were counted at all read thresholds between 0 and 10. At the read threshold of 0 reads (unfiltered results), all taxa were counted, even those with just 1 assigned read. At the read threshold of 1 read, we counted the taxa with > 1 assigned read and so on. Kaiju was not included in this test since its pre-set index includes all sequences of the dataset and cannot be excluded to render it comparable with the other methods.

In summary, the results in **Figure 1** demonstrate a case where all species present in the datasets are also present in the reference. In **Figure 2**, 16 out of the 69 species present in the dataset are missing from the reference. In **Figure 3**, 19 out of the 62 species present in the dataset are missing from the reference. In **Figure 4**, 2 out of the 13 species present in the dataset are missing from the reference and finally, in **Figure 5**, 9 out of the 20 species present in the dataset are missing from the reference.

2) The dataset used to illustrate a working use of AGMEMNON presented in supplemental is relatively trivial. This has been adequately addressed.

We would like to thank the Reviewer for the positive remark.

3) The comparator tools should also include other taxonomic predictors such as Kaiju, three of the tools currently presented are iterations of similar algorithms

This has been addressed for the most part, but note comments below

We refer the Reviewer to our detailed response in question 4 below.

4) It is considered essential that the authors compare their pipeline with HUMAnN3

The host RNA/DNA section appears to describe an attempt to separate bacterial from host reads. agagemnon is compared to a GATK Pathseq pipeline. It is not clear why the Pathseq pipeline was selected. What was the reason for not including HUMAnN3 (i.e., using the associated kneaddata pipeline) in these analyses?

We thank the Reviewer for suggesting an additional scenario for the benchmarks section. AGEMEMNON's host RNA/DNA-Seq mode has been specifically optimized for host tissue/biofluid microbiome studies and as shown in the relevant benchmarks it is highly accurate. These analyses are highly important, since they can enable the accurate quantification of intratissue microbiota, or to permit microbial analyses for studies without a relevant arm. To the best of our knowledge, GATK Pathseq was the only relevant implementation when preparing our manuscript and also is known to be a specialized application specifically for these types of analyses. Kneaddata is a relatively new (and unpublished) pipeline whose main goal, even though closely related, is different (focuses on read host/microbe tagging and sample QC).

Following the Reviewer's suggestion, we included the HUMAnN3 pipeline in our tests and applied kneaddata. HUMAnN3, following the pre-processing with kneaddata, utilizes MetaPhlAn 3 for abundance estimation. As shown in **Figure 6** below, AGEMEMNON and GATK PathSeq outperform HUMAnN3, with AGEMEMNON being the top performer. The low FP number for Kneaddata + MetaPhlAn 3 is technically deflated, due to the pre-defined very small reference query space it uses compared to AGEMEMNON and GATK Pathseq.

Figure 6 (Fig. 6 in revised manuscript): The Mean Squared Log Error (MSLE) and the number of false positive taxa (FP) between true and estimated read counts at the levels of Genus, Species and Strain using Mixed datasets one and two and the human-subset reference. We measured MSLE and False positive taxa at read thresholds between 0 and 300 with a step of 5 reads. At the read threshold of 0 reads (unfiltered results), all taxa were counted, even those with just 1 assigned read. At the read threshold of 5 reads, we counted the taxa with > 5 assigned reads and the taxa that had < 5 reads assigned were not taken into consideration and so on. Smaller MSLE and smaller numbers of false positives denote better performance. Kneaddata + MetaPhlan 3 produce results up to the species level and thus they were not included in the strain-level comparisons.

5) *Figure 1 needs to be refocused on AGAMEMNON itself to illustrate what the algorithm is actually doing*

Figure 1 is much improved, but the schematic on EM could be improved to provide more detail on what is actually happening under the hood

We thank the Reviewer for the suggestion. Since the purpose of Figure 1 is to provide to the Readers a higher-level overview of AGAMEMNON, we designed an additional figure demonstrating AGAMEMNON's quantification engine (including the EM algorithm) schematically in greater algorithmic detail. We also made a small modification in Figure 1, inside the quantification box, where we changed the text of the first box to read "Post-filtering step" and moved it out of the iterative process and before the results box. The newly created figure has also been added below:

Figure 7 (Fig. 2 in revised manuscript): Schematic representation of AGAMEMNON's quantification engine. Each black line indicates a microbial genome. In this example, most reads are unambiguously aligned to a single genome (shown as short green lines), while 6 reads map to multiple genomes (rounded red, turquoise, purple, orange, grey and yellow boxes). Each EM step consists of K iterations (default $k = 10$). In the first EM step and first iteration, multi-mapping reads are equally partially assigned to all the genomes that they align against. For example, the turquoise read that maps to three genomes, G2, G3 and G4, is assigned a base coverage/probability of 0.33 in each (shown by the same opacity of color in EM Step, first iteration). During EM, read assignments are resolved through iterations of reassigning the reads based on the abundance of the genomes/strains observed in the previous iteration. In each iteration, the quantification of each strain, as estimated based on the current read assignment, is used as the prior for multi-mapping read assignment in the subsequent iteration. Following each EM step (i.e., K iterations), the Set-cover Step is also adopted, in order to resolve special multi-mapping cases that are unsolvable by the EM, called "multi-mapping islands". These are groups of highly similar strains with low abundance for which all reads are multi-mapped making it infeasible for EM to prioritize one strain over another, leading to reporting the whole group of strains with small abundances, while only few of them exist in the sample of interest, introducing false positives. The EM step - Set-cover step is a looping process until Set-cover is unable to remove any further genomes in which case, EM process iterates until termination. In the last step of the procedure, all the genomes with abundance values lower than a predefined cutoff are removed. In the figure's example, the process starts with six genomes (G1 – G6). Throughout the iterations of the first EM step, the read probabilities change but all six genomes remain in the quantification process. When the first EM step is over, the model continues with the first Set-cover step. In the Set-cover step, only the genomes in which all reads are multi-mapped will be taken into consideration (i.e., G4, G5, G6). Through the Set-cover process, we will keep only genome G4 and remove genomes G5 and G6 aiming for minimum number of strains that explain all multi-mapping reads. In the second EM step (not shown in the figure), only genomes G1-G4 will participate in the process. Subsequently, in this particular example, the Set-cover step will never be called again because there are no multi-mapping islands left in the reference. Thus, the EM process will iterate until termination. Finally, after the whole EM process is done, the heuristic removal step will further remove the genomes whose abundance is equal to or less than 2 reads and thus, in this example, genome G1 will also be removed before reporting the final quantification results.

6) The EM method needs to be more clearly explained in results

The attempts to clarify are appreciated. However, there is still not enough clarity for the average user (and general audience of Genome Biology) of the tool to understand the implementation of the EM algorithm. The current explanation appears relatively brief and needs to be revisited and passed by some colleagues in

neighboring labs to ensure that it is clear how the algorithm is working. More detail in schematic 1 would clearly help.

Apart from the newly introduced figure that visually explains the quantification engine, we significantly expanded the relevant text in both the Results and the Methods sections. We now provide an overview of how the EM algorithm approach the problem, the aim, the iterative process, and convergence criteria of the algorithm. We believe that the significantly enhanced description will maximize the readability of the passage.

The Reviewer can find the updated text for both the Results and the Methods sections below.

AGAMEMNON's quantification engine (Results section)

AGAMEMNON uses an expectation maximization (EM) algorithm to probabilistically resolve the origin of reads (Methods, "Quantification of microbial fragments") to individual references in the second step of the pipeline called "Cedar" (Fig. 1, Fig. 2). This step contributes to its enhanced quantification accuracy at the species and strain levels. Unlike methods such as Kraken[16] and Kraken 2[36] that propagate reads that have multiple best assignments to a higher taxonomic rank, AGAMEMNON, via the inference performed in Cedar, makes use of other reads and their probabilistic allocations to determine the probability that the ambiguous read arises from the different references to which it aligns well. Similar to other EM algorithms, in Cedar, we iterate over these two steps of Expectation and Maximization until the convergence criteria are met. In each iteration, we calculate the read probability distribution in the Expectation step and assign the reads across strains to maximize the probability of observed reads in the Maximization step. Based on the user's request, the same procedure can be applied at different levels of the taxonomy tree.

Cedar's EM procedure is specifically modified according to fundamental properties and challenges of metagenomic quantification. For instance, in metagenomic indexes there is often high similarity among the strain sequences belonging to the same species[31] (sometimes even across species), which increases the complexity of disentangling reads at lower levels of the taxonomy. Additionally, reads coming from unknown species or unknown strains can be falsely assigned to entries existing in the index, resulting in false positive non-zero values. As part of the Cedar pipeline, we address these challenges through iterative, mass-preserving filtering. We look for groups of references that share the same class of reads and are fully ambiguous without any preference towards a reference over the others to detangle reads in the Maximization step. We call such a group of references, a "multi-mapping island". We reduce the problem of multi-mapping islands to a "set-cover" problem and solve it by adopting an existing approximate set-cover solution. Essentially, we select minimum number of strains that explain all multi-mapping reads distributed across the strains of each multi-mapping island. The remaining strains in each island are removed prior to the next EM step, significantly improving the accuracy of the proposed quantification model. Through this approach, we tackle the problem by sparsifying the solution (i.e., the set of species that may be assigned a non-zero abundance) in a manner that still retains all mapped reads. The "Set-cover" step is called after every k iterations of EM until there are no multi-mapping islands left. At this point EM continues until termination, which happens either if (a) it reaches the maximum number of iterations (default = 1,000 iterations) or (b) the genomes abundance change between the two iterations is adequately small. Cedar procedure is completed by the final step of removing genomes with abundance values lower than a cutoff threshold (default = 2).

Figure 7 above shows the quantification procedure and its figure legend enables the Readers to follow an example which explains the process. The following text was added to the Methods section:

Quantification of microbial fragments (Methods section)

We model the problem of estimating the expression of microbial strains by adopting the generative model introduced in RSEM [65] for quantification of RNA-seq reads. In that model, each fragment is generated by first selecting a reference sequence and then a position on the reference. Therefore, if we assume the same model for generating microbial reads, we can write the likelihood of observing the set of fragments given a distribution for the strain expressions as follows:

$$\mathcal{L}(\theta; \mathcal{F}) = \prod_{f_j \in \mathcal{F}} \sum_{i=1}^M \Pr(r_i | \theta) \Pr(f_j | r_i)$$

Where M is number of the strains, \mathcal{F} is set of all fragments in the sample and θ is the parameter showing the strains expression estimation, $\Pr(r_i | \theta)$ is the prior probability of selecting strain r_i and $\Pr(f_j | r_i)$ is the conditional probability of generating fragment f_j from reference r_i .

To compute $\Pr(f_j | r_i)$, we evaluate the compatibility of fragment f_j with the reference r_i by considering the alignment score computed by PuffAligner[65]. Furthermore, we also consider the effective length and coverage ratio of the reference for computing the prior probability of selecting a strain to be sequenced.

In this step of the pipeline, we use the expectation maximization (EM) algorithm for estimating θ , abundances at the level of strains. We apply this iterative algorithm over a reduced representation of the data provided by rich equivalence classes[66] until the parameter estimates converge or we reach the maximum allowed number of iterations. This reduction makes the optimization of the objective function practically tractable by decreasing the amount of computation for the large space of fragment associated variables. As per definition, two fragments f_i and f_j are equivalent if they align to the exact same set of references in which case, they belong to the equivalence class that is labeled by those references. In the new optimization process, the likelihood objective function that is optimized is the probability of observing the set of equivalence classes rather than read fragments with the frequency of each equivalence class defined by total number of reads belonging to that class. Subsequently, we adjust the objective function for different biases in the data such as variation of coverage for references belonging to the same equivalence class, an identical process to the one followed in Salmon[66], as well as by applying the range factorization improvement to break the equivalence classes into more fine-tuned and accurate approximations of the fragment distributions without the loss of tractability and performance, as explained in Zakeri et al.[67]. The likelihood function after representing all the fragments as range factorized equivalent classes is:

$$\mathcal{L}(\theta; \mathcal{F}) \sim \prod_{\mathcal{F}^q \in \mathcal{C}} \left(\sum_{r_i \in \Omega(\mathcal{F}^q)} \Pr(r_i | \theta) \Pr(f | \mathcal{F}^q, r_i) \right)^{N^q}$$

Where \mathcal{C} is the set of all equivalent classes, $\Omega(\mathcal{F}^q)$ is the set of all strains in equivalent class \mathcal{F}^q , N^q is the number of fragments in the equivalent class \mathcal{F}^q , and $(f | \mathcal{F}^q, r_i)$ is the conditional probability of observing a fragment from strain r_i in equivalent class \mathcal{F}^q . As shown by the equation, we treat all the N^q fragments in the equivalent class \mathcal{F}^q equally and assign the same conditional probabilities to them given each strain. Summarizing the fragments into equivalent classes reduces the number of updates required in each EM iteration and therefore improves the efficiency of the EM algorithm.

In many metagenomic quantification tasks, considering the depth of the high throughput sequencing techniques and the large amount of sequence similarity across genomes, low abundance sequences or taxa are usually discarded from the quantification results. One solution proposed to avoid reporting a large number of false positive genomes is having a post-filtering step to discard references with abundances smaller than a predefined cutoff value or discard all the multi-mapped reads[31, 64]. This post-filtering results in losing the reads that were mapped to those discarded references. However, in our pipeline, rather than discarding these reads, we keep and distribute them among the remaining references. This can help better estimate the counts of the remaining references based on the following principles:

1. Each read is supposed to be the output of a sequencing process and contains information, so discarding the read implies losing information.
2. Each read can be multi-mapped to the true origin and some other references in case of no errors in the sequencing.
3. The sequencing error rate is low and if the mapping tool allows sub-optimal mappings, we can rely on the fact that we almost always have the true genome of origin among the set of references for a multi-mapped read.

Considering these observations, we augment the EM algorithm with an iterative, mass-preserving thresholding step.

Iterative thresholding is the solution we propose to tackle the problem of sparsifying the abundance report at which we arrive without the loss of reads. Throughout the EM process, at the beginning of every k iterations (default $k = 10$), we go over the following four steps of thresholding:

1. Mark References as Potentially Removable (PR): We examine all references and mark a reference as potentially removable if it has a count smaller than or equal to the given cutoff. Our main goal is to discard as many references as possible from the list of potentially removable references without losing any reads.

2. Remove Safe PR References Immediately: A read will be lost if all the references that are equivalent over this read are discarded as the equivalence class that the read belongs to will contain no remaining references. Therefore, out of the list of the PR references, those for which there is a non-PR reference belonging to each equivalence class in which this reference appears in the label can be safely removed.

3. Solve the Set Cover Problem for Unsafe PR References: For any reference g_i in the remaining set of PR references, there exists at least one equivalence class such that all of its references are PR, including g_i . We call such equivalence classes critical equivalence classes, and according to our goal we want to determine a minimum number of PR references that can cover all critical equivalence classes, while still assuring that each aligned read can still be allocated to a retained reference. The problem can be easily reduced to set cover. Each reference represents a set of critical equivalence classes (those that it is part of) and we want to select the minimum number of sets (i.e., references) that can explain all the set members (i.e., critical equivalence classes). Set cover is NP-Hard[68], therefore, we employ the greedy approximation algorithm to obtain a set of retained references. The remaining elements (i.e., genomes) at the end of the set-cover process are those that cannot be removed. For the rest of the list, we can safely remove them without losing any reads.

4. Update Equivalence Classes: In the last step, we need to update all equivalence classes that have lost any reference and the weights of the references for the next iterations of EM.

We mention that although modifying the list of genomes through iterative thresholding, therefore, introducing a new likelihood function at each EM step, the EM algorithm's termination is still guaranteed. When we perform a set cover step, either it is idempotent, and so the set of references remains the same, and the termination of the procedure follows from the termination of the EM, or we remove at least one reference. But we can remove a reference at most N times (N= number of references). So, this procedure must always terminate.

7) The shinyapp visualization for the results is interesting and it was surprising to see this denigrated to supplemental methods.

The figure provided was great! and will be helpful in getting new users to adopt the tool

We would like to thank the Reviewer for the comment. We agree that the additional figure will enable users quickly appreciate the user-friendliness of the graphics user interface, which sets apart AGAMEMNON from the common scenario of command line or script-based interaction with such methods.

8) Documentation on the tool is quite limited

This has been adequately addressed

We would like to thank the Reviewer for the positive remarks.

9) The benchmarking assessments were quite limited and also non-intuitive, the mean square log error needs explaining better and in Figure 2, I had no idea what the two axes represent. Readers might also expect to see more traditional bar charts of taxa predicted and their relative abundances, and also, where there is a gold standard, some clearly representation of what % of which taxa were correctly predicted by each tool.

The explanation of MLSE needs to be added to the Results section. I also agree with Reviewer 1 that benchmarking would benefit from additional measures of performance. As suggested previously it would be helpful for the more general readers of Genome Biology to be presented with the more traditional bar charts of taxon predictions along with some graphic showing the accuracy of assignments. As the response to both reviewers to questions over the MLSE shows, its use is relatively abstract and likely to confuse the readers who would benefit from the addition of more straightforward graphical analyses.

By following the Reviewer's apt remark, we have added an explanation of MSLE in the Results section and also a more detailed version in the Methods section.

Calculating error-based metrics to render accuracy is pretty common in the metagenomics setting. For example, in Bracken, they calculate the *Average Error* of the true versus estimated values. In the meta-Kallisto paper, they calculate the *Average Relative Error* and the *Relative Root Mean Square Error* between actual and estimated abundances. In the MetaPhlAn 2 publication, they use the *Root Mean Squared Error* and the *Absolute Error in Relative abundance* metrics.

The *Mean Squared Log Error* metric is practically the total error between the actual and estimated values projected in the log-space (depicting errors accurately for genera/species/strains with very high and low abundances in the same plot) and normalized by the total number of reported taxa. Using this metric, we calculate the error of each method and by definition, the method with the lowest total error has the better accuracy, since it diverges the least from the true abundances.

As the methods mentioned above, we prioritized a continuous error metric, since the focus of these methods is on abundance and not presence/absence. An entry present but with a dramatically erroneous abundance will still be identified as correct if only the presence and not the distance to the ground truth is taken into account. However, for completeness and because often such methods are plagued by hundreds or thousands of species having very small numbers of false positive reads assigned to them, we added the number of false positive taxa reported by each method. This is especially useful for most of the researchers working in the fields of environmental metagenomics, human microbiome etc. especially if they conduct diversity index-based analyses where the number of identified genera, species etc. can dramatically alter the final results.

10) The results reporting strain (and even species) assignment accuracy were a little perplexing.

For the Illumina400 dataset it is not clear what the authors mean when they state they removed 63% of the strain sequences, ideally it would be the entire genome (or complete set of sequences for a specific strain) that were removed, but it seems that instead the authors randomly removed ~2/3 of each of the 400 strains sequences present in the dataset - is this correct? If the entire genome was removed, then it is not clear how Figure 2 can provide metrics for the performance of the tools at strain level (since all the reference strain sequences should have been removed) - maybe I am missing something here, but as presented this is not reflecting what happens when you don't have the strain you are searching for in your reference database.

In all cases entire genomes are removed. By following the Reviewer's suggestion, we have added relevant clarifications in the section describing the construction of the microbial references. The text now reads:

*"The Illumina 400 simulated dataset comprises sequencing reads originating from 400 microbial genomes. Reference 3 (REF-3) consists of ~8,600 microbial genomes. We removed 63% of all genomes (i.e., 252 entire genomes) from reference 3 that are part of the simulated dataset. After the aforementioned removal, reference 3 contains only 148 out of the 400 genomes that are part of the illumina 400 dataset. Strain level results for that particular scenario are calculated by taking into account **(a)** how accurate the quantification of the abundances is for the 148 strains that are both part of the reference and present in the dataset and **(b)** how many reads are mis-classified into different strains for the 252 strains that are missing from the reference but are part of the dataset. In the scenario where 100% of the strains present in the dataset are missing from the reference, we believe it is still informative to calculate metrics of accuracy. In that case, the number of the false positive strains represents the number of falsely reported taxa (in terms of presence/absence) and MSLE shows the total error in terms of mis-assigned counts (i.e., the degree of overestimated abundances for each of the falsely reported taxa). For example, a method that does not assign reads of missing strains to those present in the index will outperform a method assigning falsely the majority of these reads (while keeping all other assignments equal)."*

Regarding the authors response:

"Regarding BWA alignment per genome, reads could map on multiple genomes (sequences are extensively shared across strains and species) and the final number of high-quality mappings will be vastly larger than the number of original reads."

This would actually be good to demonstrate in your benchmarking

By following the Reviewer's suggestion, we also compared AGAMEMNON with BWA. As demonstrated in **Figures 8 and 9** below, AGAMEMNON outperforms BWA in all test sets and metrics.

BWA proved to be a technical challenge, since for the indexing step of reference 3, BWA required **~7 days**. We believe that the 7 days requirement for the indexing step is unpractically time- and resource-consuming but to be fair, BWA was not designed for the purpose of analyzing metagenomics datasets. This is the reason why the direct transfer of technology used in standard NGS analyses to the metagenomics field fails most of the times.

Figure 8: The Mean Squared Log Error (MSLE) and the number of false positive taxa (FP) between true and estimated read counts at the levels of Genus, Species and Strain using the Illumina 400 dataset and **reference 1**. We measured MSLE (a) using unfiltered results (0 x axis tick) and (b) by removing all instances where the true and estimated counts were both zero (1 x axis tick). False positive taxa were counted at all read thresholds between 0 and 10. At the read threshold of 0 reads (unfiltered results), all taxa were counted, even those with just 1 assigned read. At the read threshold of 1 read, we counted the taxa with > 1 assigned read and so on.

Figure 9: The Mean Squared Log Error (MSLE) and the number of false positive taxa (FP) between true and estimated read counts at the levels of Genus, Species and Strain using the Illumina 400 dataset and **reference 3**. We measured MSLE (a) using unfiltered results (0 x axis tick) and (b) by removing all instances where the true and estimated counts were both zero (1 x axis tick). False positive taxa were counted at all read thresholds between 0 and 10. At the read threshold of 0 reads (unfiltered results), all taxa were counted, even those with just 1 assigned read. At the read threshold of 1 read, we counted the taxa with > 1 assigned read and so on.

11. The emphasis on host filtering is a little misplaced

See comment 4 above

We refer the Reviewer to our detailed response in question 4.

12. In Figure 4 Gardnerella was reported in Peyers patch

The references cited were not convincing, largely reliant on 16S rDNA surveys, although the Front Nutr study did suggest that they confirmed with qPCR, although that may have been confounded by the choice of primers used. Given lack of wider supporting literature, these may therefore represent mis-annotations. It would be important to provide some sort of note explaining this finding, with these caveats, in the author's own data.

We strongly agree with the Reviewer that all these findings could potentially be incidental, even though they were identified in diverse publications. We have added a clear "caveat emptor" along with the references in order to provide to the Readers an objective and informed description. The Reviewer can find the added text below:

"Most of the identified species are known to be present in the human gut. Gardnerella Vaginalis is a rather surprising finding, which following literature evaluation it has also been reported in Zhou et al. (Front Nutr 2021) and Gopalakrishnan et al. (Science, 2018). However, since the ground truth in these datasets is not known it is not possible to evaluate further the accuracy of these quantifications."

13. The single cell results were not well described

This has been adequately addressed

We would like to thank the Reviewer for the positive remark.

Third round of review

Reviewer 2:

The authors have responded well to the previous round of comments, there are a few minor outstanding items that should be resolved in consultation with the Editor.

Regarding comment 4, it was suggested that the authors compare their pipeline with kneaddata to compare their relative ability to filter for host reads (the suggestion previously being that filtering out host reads through e.g. BWA searches against the host genome is relatively trivial). I apologize if I was previously not clear in my previous comments, but what the authors have apparently done is compare these pipelines in the context of genus/species assignments for Microbial Taxa. The authors should present an analysis of host read filtering i.e. simply report how many reads were correctly assigned to host by the three tools.

Regarding comment 5, the new figure is very informative, I would suggest reducing the number of multimapped reads as the colour gradients get a little bit lost in the EM steps

Regarding comment 10, the authors report the number of false positive taxa in Figure 3, predicted by different tools. To further clarify what is being presented here, could the authors also show the number of reads that have been misclassified at each of the three levels of taxonomy (can be combined in a single graphic as obviously there is no read threshold required). This gets at the idea that one tool might misassign a few reads associated with a lot of taxa as opposed to a lot of reads associated with fewer taxa. This information is currently a little bit obscured as currently presented in Figure 3. Again I may be missing something here and this may be captured by MLSE, but the additional graph would certainly reduce confusion.

Response to Reviewer 2

Great job responding to the previous round of comments, I provide only minor comments that I think might help clarify some areas I had a little bit of confusion with.

The authors have responded well to the previous round of comments, there are a few minor outstanding items that should be resolved in consultation with the Editor.

We would like to thank the Reviewer for the insightful and highly constructive comments. We believe that the additional comments and suggestions helped towards strengthening our manuscript and the way the method and results are presented.

Regarding comment 4, it was suggested that the authors compare their pipeline with kneaddata to compare their relative ability to filter for host reads (the suggestion previously being that filtering out host reads through e.g., BWA searches against the host genome is relatively trivial). I apologize if I was previously not clear in my previous comments, but what the authors have apparently done is compare these pipelines in the context of genus/species assignments for Microbial Taxa. The authors should present an analysis of host read filtering i.e., simply report how many reads were correctly assigned to host by the three tools.

We appreciate the Reviewer's comment. AGAMEMNON's host RNA/DNA-Seq mode is a comprehensive analysis approach, going beyond host read filtering. Starting with raw RNA/DNA-Seq host tissue-specific samples, AGAMEMNON produces **(a)** the BAM files comprising alignment results for the host reads which can directly be used for downstream analyses and **(b)** the files that contain the microbial quantification results. Such analyses can be useful in numerous scenarios, from contaminant identification and quantification to combined host-microbiome/viral analysis in healthy or diseased tissues.

Regarding the comparisons, kneaddata labels sequencing reads as "host-derived" and "microbiome-derived" by producing separate FASTQ files comprising the "host" and the "microbiome" reads, while GATK Pathseq only attempts to identify and quantify microbial abundances in such samples and does not produce any results for the host reads (e.g., SAM/BAM/FASTQ files). Thus, for this particular analysis, we proceeded with AGAMEMNON, Kneaddata and also BWA as a baseline. We applied the algorithms on a dataset with ~62M host and ~2.5M microbiome derived reads (~3.9% microbial reads), mentioned in the manuscript as dataset "Mixed one" and also a dataset with ~62M host and ~5M microbiome derived reads, mentioned in the manuscript as dataset "Mixed two" (~7.5% microbial reads). We created a table (**Additional file 4: Supplementary Table S14**) showing the number of correctly- and mis-aligned reads to the host and the microbiome.

As shown in **Additional file 4: Supplementary Table S14**, even though all three methods produced similar numbers of correctly aligned reads to the host, Kneaddata also misaligned 32,382 and 218,685 reads, while BWA misaligned 388 and 1,688 reads of bacterial origin to the host in datasets one and two respectively, while the assignment performed by AGAMEMNON has 0 misaligned reads in both cases. Importantly, HISAT2 which is utilized by AGAMEMNON can also handle host-derived RNA-Seq samples by aligning reads to the host genome in a splice-aware manner. On the contrary, GATK Pathseq

and Kneaddata (bowtie2) as well as BWA are designed for the analysis of only DNA-Seq datasets and can miss reads spanning splice junctions. This is a known weakness when DNA aligners are used for RNA-Seq data.

The two simulated datasets created in this study are both DNA-Seq in order to render the results directly comparable between the 3 algorithms. AGAMEMNON has the highest number of correctly assigned and the lowest number of falsely assigned microbiome reads in almost all of the cases.

Method	Dataset	Correct host reads (%)	Misaligned reads (host)	Correct microbiome reads (Genus)	Misassigned microbiome reads (Genus)	Correct microbiome reads (Species)	Misassigned microbiome reads (Species)
Ground truth	Mixed one	61,913,525 - (100%)	-	-	-	-	-
AGAMEMNON		61,378,400 - (99.14%)	0	2,252,420	180,202	1,739,614	693,008
Kneaddata		61,828,553 - (99.86%)	32,382	1,355,901	324,683	1,355,902	324,683
GATK Pathseq		-	-	2,010,620	4,414,403	1,797,472	7,261,250
BWA		61,903,489 - (99.98%)	388	-	-	-	-
Ground truth	Mixed two	61,910,514 - (100%)	-	-	-	-	-
AGAMEMNON		61,373,945 - (99.13%)	0	4,787,912	133,475	4,400,468	520,927
Kneaddata		61,684,173 - (99.63%)	218,685	2,198,500	539,839	2,198,500	539,839
GATK Pathseq		-	-	3,866,137	8,272,749	3,866,137	10,749,569
BWA		61,900,521 - (99.98%)	1,688	-	-	-	-

Regarding comment 5, the new figure is very informative, I would suggest reducing the number of multimapped reads as the colour gradients get a little bit lost in the EM steps.

We thank the Reviewer for the suggestion. We have removed one of the two cases where a read has three multimapping positions and made also some additional modifications to the rest of the reads (colors and positions) to make the figure easier to follow. The updated figure has also been added below:

Regarding comment 10, the authors report the number of false positive taxa in Figure 3, predicted by different tools. To further clarify what is being presented here, could the authors also show the number of reads that have been misclassified at each of the three levels of taxonomy (can be combined in a single graphic as obviously there is no read threshold required). This gets at the idea that one tool might misassign a few reads associated with a lot of taxa as opposed to a lot of reads associated with fewer taxa. This information is currently a little bit obscured as currently presented in Figure 3. Again I may be missing something here and this may be captured by MLSE, but the additional graph would certainly reduce confusion.

We thank the Reviewer for the suggestion. We created two additional figures (**Figs 1, 2**) showing **(a)** the total number of misclassified reads originating from the false positive taxa identified by each method and for each taxonomic rank and **(b)** the total number of correctly classified reads. The newly added figures are based on the following scenario (showcased in figure 3 in the original manuscript):

- Illumina 400 dataset – Reference 1 (Figure 3 in the original manuscript)

As can be seen in the figures below, at the genus and species levels, AGAMEMNON has the lowest number of misassigned reads with Kallisto following while being second best at the strain level behind Kallisto. Furthermore, AGAMEMNON has a slight advantage over all methods and taxonomic ranks in terms of total number of correctly classified reads.

Figure 1: Number of misclassified reads at the genus, species and strain levels (Illumina 400, Reference 1).

Figure 2: Number of correctly classified reads at the genus, species and strain levels (Illumina 400, Reference 1).